# 1 Carbon uptake and water use in woodlands and forests in southern

# Australia during an extreme heat wave event in the 'Angry Summer' of 2012/2013.

- Eva van Gorsel<sup>1</sup>, Sebastian Wolf<sup>2</sup>, James Cleverly<sup>3</sup>, Peter Isaac<sup>1</sup>, Vanessa Haverd<sup>1</sup>, Cäcilia Ewenz<sup>4</sup>,
- Stefan Arndt<sup>5</sup>, Jason Beringer<sup>6</sup>, Víctor Resco de Dios<sup>7</sup>, Bradley J. Evans<sup>8</sup>, Anne Griebel<sup>5,9</sup>, Lindsay B.
- Hutley<sup>10</sup>, Trevor Keenan<sup>11</sup>, Natascha Kljun<sup>12</sup>, Craig Macfarlane<sup>13</sup>, Wayne S. Meyer<sup>14</sup>, Ian McHugh<sup>15</sup>,
- Elise Pendall<sup>9</sup>, Suzanne M. Prober<sup>13</sup>, Richard Silberstein<sup>16</sup>
- 1 CSIRO, Oceans and Atmosphere, Yarralumla, 2600, Australia
- 2 Department of Environmental Systems Science, ETH Zurich, 8092 Zurich, Switzerland
- 3 School of Life Sciences, University of Technology Sydney, Broadway, NSW, 2007, Australia
- 4 Airborne Research Australia, Flinders University, Salisbury South, SA, 5106, Australia
- 5 School of Ecosystem and Forest Sciences, The University of Melbourne, Richmond, 3121, Victoria, Australia
- 6 School of Earth and Environment (SEE), The University of Western Australia, Crawley, WA, 6009, Australia
- 7 Producció Vegetal i Ciència Forestal Agrotecnio Centre, Universitat de Lleida, 25198, Leida, Spain
- 8 School of Life and Environmental Sciences, The University of Sydney, Sydney, 2015, Australia
- 9 Hawkesbury Institute for the Environment, Western Sydney University, Penrith, NSW 2570
- 10 School of Environment, Research Institute for the Environment and Livelihoods, Charles Darwin University, NT,
   Australia
- 18 Australia19 11 Lawrence Berkeley National Lab., 1 Cyclotron Road, Berkeley CA, USA
- 20 12 Dept of Geography, College of Science, Swansea University, Singleton Park, Swansea, UK
- 21 13 CSIRO Land and Water, Private Bag 5, Floreat 6913, Western Australia
- 22 14 Environment Institute, The University of Adelaide, Adelaide SA 5 005, Australia
- 23 15 School of Earth, Atmosphere and Environment, Monash University, Clayton, 3800, Australia
- 16 Centre for Ecosystem Management, Edith Cowan University, School of Natural Sciences, Joondalup, WA, 6027,
   Australia
- Correspondence to: E. van Gorsel (eva.vangorsel@gmail.com)
- Abstract. As a result of climate change warmer temperatures are projected through the 21st century and are already
- increasing above modelled predictions. Apart from increases in the mean, warm/hot temperature extremes are expected to
- become more prevalent in the future, along with an increase in the frequency of droughts. It is crucial to better understand

the response of terrestrial ecosystems to such temperature extremes for predicting land-surface feedbacks in a changing 31 climate. While land-surface feedbacks in drought conditions and during heat waves have been reported from Europe and 32 the US, direct observations of the impact of such extremes on the carbon and water cycles in Australia have been lacking. 33 During the 2012/2013 summer, Australia experienced a record-breaking heat wave with an exceptional spatial extent that 34 lasted for several weeks. In this study we synthesised eddy-covariance measurements from seven woodlands and one 35 forest site across three biogeographic regions in southern Australia. These observations were combined with model results 36 from BIOS2 (Haverd et al., 2013, Biogeosciences, 10: 2011-2040) to investigate the effect of the summer heat wave on the carbon and water exchange of terrestrial ecosystems which are known for their resilience toward hot and dry 37 38 conditions. We found that water-limited woodland and energy-limited forest ecosystems responded differently to the heat 39 wave. During the most intense part of the heat wave, the woodlands experienced decreased latent heat flux (23% of 40 background value), increased Bowen ratio (154%) and reduced carbon uptake (60%). At the same time the forest 41 ecosystem showed increased latent heat flux (151%), reduced Bowen ratio (19%) and increased carbon uptake (112%). 42 Higher temperatures caused increased ecosystem respiration at all sites (up to 139%). During daytime all ecosystems 43 remained carbon sinks, but carbon uptake was reduced in magnitude. The number of hours during which the ecosystem 44 acted as a carbon sink was also reduced, which switched the woodlands into a carbon source on a daily average. 45 Precipitation occurred after the first, most intense part of the heat wave, and the subsequent cooler temperatures in the 46 temperate woodlands led to recovery of the carbon sink, decreased the Bowen ratio (65%) and hence increased 47 evaporative cooling. Gross primary productivity in the woodlands recovered quickly with precipitation and cooler 48 temperatures but respiration remained high. While the forest proved relatively resilient to this short-term heat extreme the 49 response of the woodlands is the first direct evidence that the carbon sinks of large areas of Australia may not be 50 sustainable in a future climate with an increased number, intensity and duration of heat waves.

#### 51 **1 Introduction**

Average temperatures in Australia have increased by 0.9°C since 1910 (CSIRO and BOM, 2014), which represents the 53 most extreme of modeling scenarios, and even further warming is projected with climate change (IPCC, 2013). In addition 54 to increased mean temperature, warm temperature extremes are becoming more frequent in Australia and worldwide 55 (Lewis and King, 2015, Steffen, 2015) and an increased prevalence of drought is expected for the future (Dai, 2013). 56 Increases in temperature variability also affect the intensity of heat waves (Schär et al., 2004). Extreme heat and drought 57 often co-occur (King et al. 2014), and soil water limitations can exacerbate the intensity of heat waves (Fischer et al., 58 2007; Seneviratne et al., 2010) due to reduced evaporative cooling and increased sensible heat flux (Sheffield et al., 2012). 59 This combination of reduced water availability and increased evaporative demand places increased stress on terrestrial ecosystems. 60

- During summer 2012/2013, Australia experienced a record-breaking heat wave that was deemed unlikely without climate change (Steffen, 2015). The Australian summer 2012/2013 was nicknamed the 'Angry Summer' or the 'Extreme 63 Summer', as an exceptionally extensive and long-lived period of high temperatures affected large parts of the continent in 64 65 late December 2012 and the first weeks of January 2013 (Bureau of Meteorology, BOM, 2013). Record temperatures were 66 observed in every Australian State and Territory, and the record for the hottest daily average temperature (32.4°C) for 67 Australia was recorded on the 8<sup>th</sup> of January (Karoly et al., 2013). On the Western Australian South Coast, the maximum temperature record was broken in Eucla on the 3<sup>rd</sup> of January with 48.2°C. In South Australia maximum temperature 68 records were broken at four weather stations between the 4<sup>th</sup> and 6<sup>th</sup> of January. Victoria also observed record heat on the 69 4<sup>th</sup> of January at its south coast in Portland (42.1°C). In New South Wales, record temperatures were recorded on the 5<sup>th</sup> of 70 January and were broken again on the 19th, reaching 46.2°C before the heat wave subsided. Besides being the hottest year 71 72 since 1910, summer 2012/2013 was also considerably drier than average in most parts of the continent, but particularly in 73 the densely populated east of Australia. King et al. (2014) have shown that extreme heat was made much more likely by 74 contributions from the very dry conditions over the inland eastern region of Australia as well as by anthropogenic 75 warming.

Heat waves are becoming hotter, they last longer, and they occur more often (Steffen, 2015). As many ecological 78 processes are more sensitive to climate extremes than to changes in the mean state (Hanson et al. 2006), it is imperative to 79 understand the effect of climate extremes in order to predict the impact on terrestrial ecosystems. Processes and 80 sensitivities differ among biomes, but forests are expected to experience the largest detrimental effects and the longest 81 recovery times from climate extremes due to their large carbon pools and fluxes (Frank et al. 2015). There is increasing 82 evidence that climate extremes may result in a decrease in carbon uptake and carbon stocks (Zhao and Running, 2010, 83 Reichstein et al., 2013). It is therefore crucial to better understand ecosystem responses to climate extremes. The role of 84 climate extremes could be critical in shaping future ecosystem dynamics (Zimmermann et al. 2009), but the sporadic and 85 unpredictable nature of these events makes it difficult to monitor how they affect vegetation through space and time 86 (Mitchell et al., 2014).

Australian forest and woodland ecosystems are strongly influenced by large climatic variability, characterised by recurring 89 drought events and heat waves (Beringer et al., 2016, Mitchell et al., 2014). Eucalyptus regnans ecosystems in southeast 90 Australia, for example, have an exceptional capacity to withstand drought and the ability to recover almost instantly after 91 heat waves (Pfautsch and Adams, 2013). However, drought and heat related forest die-back events have been observed in 92 southwestern Australia (Matusick et al. 2013, Evans and Lyons 2013), where drought stress from long-term reductions in 93 rainfall have been exacerbated by short heat wave periods. This suggests that these ecosystems, even though they are 94 resilient to dry and hot conditions, are susceptible to mortality events once key thresholds have been exceeded (Evans et 95 al., 2013). Similar large-scale droughts and heat waves in Europe during 2003 (Ciais et al., 2005), in Canada during 2000

to 2003 (Kljun et al. 2007) and in the US during 2012 (Wolf et al. 2016) caused substantial reductions in summer carbon

uptake, and vegetation-climate feedbacks were found to contribute to enhanced temperatures (Teuling et al., 2010, Wolf et

al., 2016). However, direct observations of the ecosystem response to large-scale extremes in Australia have been lacking
 until very recently.

The large spatial extent of the heat wave in early 2013 across Australia and direct observations from the OzFlux network enable us for the very first time to analyse the effect of extreme hot and dry conditions on the carbon, water and energy cycles of the major woodland and forest ecosystems across southern Australia. In this study, we combined eddycovariance measurements from seven woodland and forest sites with model simulations from BIOS2 (Haverd et al. 2013) to investigate the impact of the 2012/2013 summer heat wave and drought on the carbon and water exchange of terrestrial ecosystems across climate zones in southern Australia and to assess the influence of land-surface feedbacks on the magnitude of the heat wave.

#### 108 **2 Materials and Methods**

We compared hourly data from seven OzFlux sites (Fig. 1, Table 1) measured during the heat wave period 1 January 2013 110 - 18 January 2013 to observations from a background reference. We used eddy covariance data to compare hourly data 111 and the daily cycle of latent and sensible heat as well as carbon fluxes. We used the measured hourly data of a background 112 period (BGH) one year later from 2/1/2014 - 6/1/2014. During these time periods all towers were actively taking 113 measurements, although data gaps were present after 18 January in 2013. The reference period was shorter than the heat 114 wave period because another significant heat wave event affected southeastern Australia in late January 2014 during a 115 time period when not all sites had comparable data available in 2013. Temperatures during the background reference 116 period were also somewhat warmer than average climatology (Fig. 2). We therefore expect the relative severity of the 117 effects of the heat wave to appear smaller than they otherwise would when compared against a climatological reference. 118 To ensure the representativeness of our results, we also compared daily data against a climatology derived from daily 119 BIOS2 (see below) output for the time period 1982-2013 (background climatology, BGC). BIOS2 results for the whole 120 time period were only available as daily values.

#### 121 2.1 Sites

We analysed data from seven southern Australian sites (Beringer et al., 2016) grouped in three distinct ecosystem and

- climate types: Mediterranean woodlands (MW), temperate woodlands (TW) and temperate forests (TF) (Fig. 1, Table 1).
- 125 MW sites included i) a coastal heath Banksia woodland (Gingin: AU-Gin); ii) a semi-arid eucalypt woodland dominated
- by Salmon gum (*Eucalyptus salmonophloia*), with Gimlet (*E. salubrious*) and other eucalypts (Great Western Woodlands:

AU-Gww); and iii) a semi-arid mallee ecosystem (Calperum: AU-Cpr), which is characterised by an association of mallee eucalypts (E. dumosa, E. incrassata, E. oleosa and E. socialis) and spinifex hummocks (Triodia basedowii) (Sun et al. 128 129 2015, Meyer et al., 2015). TW sites are classified as dry sclerophyll woodlands and include: i) Wombat (AU-Wom), a 130 secondary re-growth of Messmate Stringybark (E. oblique), Narrow-Leaved Peppermint (E. radiate) and Candlebark (E. 131 rubida); ii) Whroo (AU-Whr), a box woodland mainly composed of Grey Box (E. microcarpa) and Yellow Gum (E. 132 leucoxylon) with smaller numbers of Ironbark (E. sideroxylon) and Golden Wattle (Acacia pycnantha); iii) Cumberland 133 Plains (AU-Cum), where the canopy is dominated by Gum-topped Box (E. moluccana) and Red Ironbark (E. fibrosa), 134 which host an expanding population of mistletoe (Amyema miquelii). Temperate Forests (TF) are represented by the Tumbarumba site (AU-Tum), which is in a wet sclerophyll forest dominated by Alpine Ash (E. delegatensis) and 135 Mountain Gum (E. dalrympleana) (Leuning et al., 2005). 136

The sites fall into the classifications "Mediterranean Forests, Woodland and Scrub" (AU-Gin, AU-GWW and AU-Cpr) or the "Temperate Broadleaf and Mixed Forest" (AU-Wom, Au-Cum, AU-Whr and AU-Tum) classifications of IBRA 139 140 (Interim Biogeographic Regionalisation for Australia v. 7; Environment, 2012). In temperate Australia both woodlands 141 and forests are mainly dominated by Eucalyptus species. Forests occur in the higher rainfall regions and woodlands form 142 the transitional zone between forests and grass-shrublands of the drier interior. We therefore classified temperate 143 ecosystems with mean annual precipitation > 1000 mm and tree height > 30 m as forests. There was only one temperate, 144 wet sclerophyll forest for which data was available during this heat wave, but we are confident that it is representative of 145 the energy limited temperate forests of southern Australia (e.g. van Gorsel, 2013). None of the sites is continental, but 146 elevations range from 33 m asl (AU-Cum) to 1260 m asl (AU-Tum). The mean annual temperature for the years 1982 -147 2013 ranged from 9.8 °C in AU-Tum to 18.7°C in AU-Gww (Table 1). Mean annual precipitation also covered a large range from 265 mm year<sup>-1</sup> in AU-Cpr to 1417 mm year<sup>-1</sup> in AU-Tum. 148

#### 149 **2.2 OzFlux Data**

We analysed data collected by the OzFlux network (www.OzFlux.org.au). Each site has a set of eddy covariance (EC) 151 instrumentation, consisting of an infrared gas analyser (LI-7500 or LI-7500A, LI-COR, Lincoln, NE, USA) and a 3D 152 sonic anemometer (generally a CSAT3 (Campbell Scientific Instruments, Logan, UT, USA) except for AU-Tum, where a 153 Gill-HS is operational (Gill Instruments, Lymington, UK)). Supplementary meteorological observations include radiation 154 (4 component CNR4 or CNR1, Kipp and Zonen, Delft, Netherlands) and temperature and humidity (HMP45C or HMP50, 155 Vaisala, Helsinki, Finland). Soil volumetric water content was measured with CS616 (Campbell Scientific). EC data were processed using the OzFlux-QC processing tool (Isaac et al., 2016). Processing steps and corrections included outlier 156 157 removal, coordinate rotation (double rotation), frequency attenuation correction, conversion of virtual heat flux to sensible 158 heat flux, and the WPL correction (Tanner and Thurtell, 1969, Wesley, 1970, Webb et al., 1980, Schotanus et al., 1983, 159 Lee et al. 2004 and references therein). Friction velocity thresholds were calculated following the method of Barr et al.

- (2013). In Tumbarumba, where advection issues are known (van Gorsel et al., 2007, Leuning et al. 2008), only data from
- the early evening was used during nighttime hours (van Gorsel, 2009). Gaps in the meteorological time series were filled
- using alternate data sets, BIOS2 or ACCESS (Australian Community Climate and Earth-System Simulator) output (Bi et
- al., 2013) or climatologies (usually in this order of preference). Gaps in the flux time series were filled using a self-
- organising linear output model (SOLO-SOFM, Hsu et al., 2002, Abramowitz et al. 2006 and references therein). The
- OzFlux data used in this analysis are available from http://data.ozflux.org.au/portal/.

### 166 **2.3 BIOS2**

The coupled carbon and water cycles were modelled using BIOS2 (Haverd et al., 2013a; Haverd et al., 2013b) constrained 167 168 by multiple observation types, and forced using remotely sensed vegetation cover and daily AWAP meteorology (Raupach et al. 2009), downscaled to half-hourly time resolution using a weather generator. BIOS2 is a fine-spatial-169 170 resolution (0.05 degree) offline modelling environment, including a modification of the CABLE biogeochemical land surface model (Wang et al., 2011; Wang et al., 2010) incorporating the SLI soil model (Haverd and Cuntz, 2010). BIOS2 171 172 parameters are constrained and predictions are evaluated using multiple observation sets from across the Australian continent, including streamflow from 416 gauged catchments, eddy flux data (CO2 and H2O) from 14 OzFlux sites 173 (Haverd et al., 2016), litterfall data, and soil, litter and biomass carbon pools (Haverd et al., 2013a). In this work, we 174 175 updated BIOS2 to use the GIMMS3g FAPAR product (Zhu et al., 2013) instead of MODIS and AVHRR products for 176 prescribed vegetation cover (Haverd 2013b). The reference period used for BIOS2 (BGC), was 1982-2013, the period over 177 which remotely sensed data were available.

#### 178 2.4 Analyses

All data analyses were performed on Jupyter notebooks using Python 2.7.11 and the Anaconda (4.0.0) distribution by 180 Continuum Analytics. Differences between heat waves and reference periods were determined by calculating z-scores of 181 temperatures and soil water content during the relevant periods, z-scores represent the number of standard deviations an observation is above or below the mean, depending upon the sign of the z-score. These were calculated with the z-score 182 function of the scipy stats module for the period  $1^{st}$ -18<sup>th</sup> January relative to the mean across all years in the BIOS2 output 183 (1982-2013). The scipy stats functions bartlett and ttest ind were used to determine the significance of differences of a 184 185 range of variables between the background period (BGH or BGC) and the heat wave periods HW1 (1/1/2013-9/1/2013) 186 and HW2 (HW2, 10/1/2013–18/1/2013). Boxplots were created using Matplotlib.

#### 187 **2.5 Conventions**

We use the terminology and concepts as introduced by Chapin et al. (2006), where net and gross carbon uptake by

vegetation (net ecosystem production (NEP) and gross primary production (GPP)) are positive directed toward the surface

and carbon loss from the surface to the atmosphere (ecosystem respiration (ER)) is positive directed away from the

191 surface.

#### 192 **3 Results**

#### 193 **3.1 Heat wave characterisation**

The heat wave event commenced on the 25<sup>th</sup> of December 2012 with a build-up of extreme heat in the southwest of Western Australia. A high pressure system in the Great Australian Bight and a trough near the west coast directed hot easterly winds over the area (BOM, 2013). From December 31 the high pressure system started moving eastward, and it entered the Tasman Sea off eastern Australia on January 4<sup>th</sup>. The northerly winds directed very hot air into south eastern Australia. Temporary cooling was observed in the eastern states after the 8<sup>th</sup> of January, but a second high pressure system moved into the Bight in the meantime, starting a second wave of record breaking heat across the continent. The heat wave finally ended on the 19<sup>th</sup> of January, when southerly winds brought cooler air masses to southern Australia.

Figure 3 shows the meteorological conditions at the sites during the heat wave. Maximum temperatures as high as 46.3°C were accompanied by vapour pressure deficits up to 9.7 kPa. The soil water fraction was as low as 0.02 in MW but 203 204 increased to 0.05 and 0.4 at AU-Gin and AU-Gww respectively after synoptic rainfalls around the 12<sup>th</sup> of January. The same, but less pronounced, was also the case for the TW sites where soil water fractions increased from 0.10 to 0.18 after 205 206 rain. At the TF site, Au-Tum, soil water content decreased throughout the heat wave (HW) from 0.26 to 0.19. Due to intermittent precipitation events we analysed two parts of the heat wave separately: heat wave period 1 (HW1, 1<sup>st</sup>-9<sup>th</sup> of 207 January 2013) was characterised by very little precipitation (2 mm over all sites) and low soil water content. During heat 208 wave period 2 (HW2, 10<sup>th</sup>-18<sup>th</sup> of January 2013) precipitation occurred at most sites (12<sup>th</sup>-15<sup>th</sup> January 2013) and resulted 209 in increased soil water content at some sites and lower temperature anomalies at all sites than during HW1. 210

During HW1 temperatures were generally more than 1.5-2 standard deviations ( $\sigma$ ) higher than the 32-year mean of the background period (BGC) for these dates. At AU-Tum and AU-Gww z-scores exceeded +2 $\sigma$ . During HW2 all sites showed lower z-scores for temperature, but they were still more than +1 $\sigma$  higher than average background temperatures. The background period BGH, against which we compare the hourly data of the heat wave, was also warmer than average conditions during the past 30 years, but these z-scores were well below 1 for most sites.

Z-values indicate that soil water content was unusually low for the time of year. It was mostly more than one standard 219 deviation below average ( $\sigma < -1$ ), except at AU-Gww where soil water content was higher than average during HW2. All 220 sites except AU-Gin and AU-Gww had a lower z-score for soil water content during HW2 than HW1, indicating relatively 221 drier conditions with respect to the BIOS2 derived climatology despite the presence of rainfall during HW2. The

- background period BGH was generally less dry than the heat wave periods, one noteworthy exception being AU-Tum,
- which had very dry conditions  $(-2\sigma)$  in BGC during early January 2014. The z-scores indicate that high temperatures were
- more unusual than low soil water content during HW1. HW2 was both hot and dry.

#### 225 **3.2** Ecosystem response to dry and hot conditions

#### 226 **3.2.1 Energy Exchange**

Incoming and reflected short-wave radiation were significantly increased by only 70 Wm<sup>-2</sup> and 3 Wm<sup>-2</sup> respectively in the 227 228 energy limited ecosystem AU-Tum during the first period of the heat wave (Fig. 4, Table 2). Otherwise they remained 229 approximately the same as BGH values except at the MW sites where they were significantly reduced (by -62 Wm<sup>-2</sup>) during HW2 (Table 2). The relatively short duration of the extreme heat wave did not result in changes to albedo (not 230 shown). A warmer atmosphere and potentially increased cloud cover led to a 38 Wm<sup>-2</sup> increase in longwave downward 231 232 radiation in Western Australia. Due to increased surface temperatures, longwave radiation emitted at the land surface was significantly increased at all sites for both heat wave periods (28 Wm<sup>-2</sup> on average), but more so during HW1 (41 Wm<sup>-2</sup> on 233 average). Net radiation was significantly reduced during HW2, but only at MW sites (-35 Wm<sup>-2</sup>). At all other sites, net 234 radiation was approximately the same during HW1, HW2 and BGH. Available energy (not shown), the energy available 235 to the turbulent heat fluxes, was significantly reduced at MW and TW sites during HW1 (by 25Wm<sup>-2</sup> and 24Wm<sup>-2</sup> 236 respectively) but was about the same for HW2. It was also about the same during HW1 and HW2 at the TF site. 237

Figure 5 demonstrates how remarkably different the energy partitioning was at MW, TW and TF sites, as we would expect given their large climatological and biogeographic differences (Beringer et al., 2016). While similar fractions of energy went into latent and sensible heat at the TF site, more energy was directed into sensible heat at TW sites. This energy flux partitioning toward sensible heat was more pronounced at MW sites, where both the mean and the variability of latent heat flux were very small due to severe water limitations. Most of the available energy was transferred as sensible heat and hence contributed to the warming of the atmosphere which was also observed for BGH.

During HW1, the generally small latent heat flux at the MW sites (38 Wm<sup>-2</sup>) was further reduced by -12 Wm<sup>-2</sup> (Table 2). During HW2, precipitation temporarily increased water availability, returning latent heat flux to levels observed during BGH. Latent heat flux did not change significantly at TW sites during the HWs compared to BGH conditions. At TF, however, latent heat flux increased by 52 and 14 Wm<sup>-2</sup> during HW1 and HW2 respectively. This was partly due to the very dry conditions in the background period BGH, but daily latent heat flux was also increased compared to the climatology (BGC, Fig. A2), particularly during HW1.

- With values exceeding 7, the observed ratio of sensible to latent heat, the Bowen ratio ( $\beta$ , Bowen, 1926), was very large in 253 254 the Mediterranean woodlands (Fig. 6). Typical values for βreach 6 for semi-arid to desert areas (e.g. Oliver, 1987, 255 Beringer and Tapper, 2000). During the heat wave these values were larger than 10. With rainfall and increased latent heat flux,  $\beta$  decreased to below background conditions in HW2 (6.4) across the MW sites. At TW,  $\beta$  was higher than 256 257 background values during HW1 (reaching a maximum value of 4.0) but decreased to background values during HW2 (2.8). For the TF site,  $\beta$  was lower (0.7 and 0.8 during HW1 and HW2 respectively) than during the background period 258 259 (1.0). It increased steadily in the morning, declined toward the evening and was quite symmetric, while in TW  $\beta$  increased 260 strongly in the afternoon during the heat waves. This increase of  $\beta$  toward the afternoon hours was observed in MW 261 during all time periods (including BGH).

Measured daily latent heat fluxes and  $\beta$  were consistent with flux climatology derived from BIOS2 during the background 264 (BGC) (Fig. A1).

#### 265 **3.2.2 Carbon Exchange**

Patterns of carbon fluxes were similar to between-site patterns of energy fluxes (Fig. 7, note differences in y-axes). All 266 267 sites showed that maximum carbon uptake (GPP) occurred in the morning, decreased throughout the afternoon, and mostly increased again in the late afternoon. NEP followed the diurnal course of GPP, with the offset related to total 268 ecosystem respiration (ER). ER increased with temperature and reached a maximum in the early afternoon (not shown). 269 Maximum NEP at MW decreased from 4.16 µmol m<sup>-2</sup> s<sup>-1</sup> during background conditions to 2.2 µmol m<sup>-2</sup> s<sup>-1</sup> in HW1 and 270 271 3.3 µmol m<sup>-2</sup> s<sup>-1</sup> in HW2. Not only did the total amount of carbon uptake decrease, but the number of hours during which 272 the ecosystem was sequestering carbon also decreased from 11.5 hours in background conditions to 10.5 during HW1 and 273 9.0 in HW2. The same was true in TW and TF in that maximum NEP was lower during the heat wave periods and the 274 time during which the ecosystems acted as sinks was shortened.

Carbon uptake was significantly reduced at MW and TW during HW1 (Fig. 8) with daytime averages decreasing from 4.6 276 µmol m<sup>-2</sup> s<sup>-1</sup> to 3.1 in MW and from 11.2 to 6.2 µmol m<sup>-2</sup> s<sup>-1</sup> in TW. In TF, however, carbon uptake was increased from 277 24.2 µmol m<sup>-2</sup> s<sup>-1</sup> to 26.5 µmol m<sup>-2</sup> s<sup>-1</sup> during HW1 and to 27.0 during HW2. Ecosystem respiration increased significantly 278 279 in both periods of the heat wave and across all ecosystems. Consequently, NEP was significantly reduced at MW and TW 280 sites during both heat wave periods, unchanged at the TF site during HW1, but increased at TF during HW2. During 281 daytime all ecosystems remained carbon sinks during the event but as there were fewer hours and decreased carbon uptake 282 during the day the woodlands switched into carbon sources. Precipitation after HW1 and cooler temperatures during HW2 283 led to a recovery of the carbon sink in TW during HW2. TF was a strong sink of carbon and remained so during both HW 284 periods.

Measured GPP and ER showed the same responses in carbon uptake and losses during the heat waves as the flux

climatology derived with BIOS2 (BGC, Fig. A2): GPP was reduced during HW1 in woodland ecosystems and increased in the forest during both heat wave periods. ER was increased at all sites and during HW1 and HW2 compared to the long-

term climatology.

#### 290 4 Discussion

#### 291 4.1 Consequences of Australian heat waves on energy fluxes

Persistent anticyclonic conditions during the 'Angry Summer of 2012/13' led to a heat wave by transporting warm air 293 from the interior of the continent to southern Australia. Such synoptic conditions are the most common weather pattern 294 associated with Australian heat waves (Steffen et al., 2014). However, these weather patterns did not result in increased 295 amounts of available energy at the surface, which was in contrast to heat waves observed in Europe and the US (see 296 section 4.4). Instead, in our study the energy available for turbulent heat fluxes was similar to or even smaller than 297 background conditions. Background conditions over Australia tend to have large available energy fluxes, even during very cyclonic periods (e.g., the 2010-2011 fluvial; Cleverly et al. 2013). Thus, differences in latent and sensible heat fluxes at 298 299 the Australian sites used in this study were due to anomalous temperature and soil moisture content rather than to changes 300 in available energy.

During the heat wave, available energy preferentially increased sensible heat flux and led to a subsequent increase of  $\beta$  at 303 drier sites (MW and TW) while at the TF site, available energy preferentially increased latent heat flux. The diurnal cycle 304 of  $\beta$  at the MW sites generally showed an increase of  $\beta$  toward the afternoon hours. This increase was more pronounced 305 during the heat wave periods than during BGH, indicating stress-induced reduction of stomatal conductance (Cowan and 306 Farquhar 1977). At TW sites,  $\beta$  only had a pronounced asymmetry during heat waves, clearly showing stronger stomatal control than during background conditions. At the TF site,  $\beta$  was lower during heat waves, but the symmetry in  $\beta$  indicates 307 308 that a decrease in midday stomatal conductance was either counteracted by increased soil evaporation under a steadily 309 increasing humidity deficit with rising temperatures from morning to mid-afternoon (Tuzet et al. 2003), or that there was 310 little stomatal control of the latent heat flux at this site, or a combination of both. Stomatal closure and the associated 311 partitioning of available energy is important as an increased  $\beta$  in response to heat waves (MW and TW) promotes further heating of the atmosphere, whereas increased latent heat flux suppresses further atmospheric heating (Teuling et al., 312 313 2010). This is only possible as long as the latent heat flux is not limited by soil water, particularly during the period of 314 peak insolation (Wolf et al. 2016). At TF the relative extractable water was above a threshold of 0.4 (J. Suzuki, pers. 315 comm.) for all but the last two days of the heat wave (not shown), indicating that for most of the time soil water was not

- limiting the latent heat flux (Granier et al., 1999). Thus, evaporative cooling from latent heat suppressed further heating
- but depleted soil moisture at the TF site. Eventually, depleted soil water stores can lead to a positive (enhancing) feedback
- on temperatures as more energy goes into the sensible than the latent heat flux further amplifying heat extremes by
- biosphere–atmosphere feedback (Whan et al. 2015). Indeed, the data indicate that toward the end of the heat wave, such
- positive feedbacks had shifted energy partitioning toward sensible heat flux at all sites.

# 321 **4.2 Impact of heat waves on carbon fluxes**

Heat waves and drought can affect photosynthesis (Frank et al., 2015). By means of stomatal regulation, plants exert 323 different strategies to balance the risks of carbon starvation and hydrological failure (Choat et al. 2012). These strategies 324 particularly come into play during extreme events (Anderegg et al. 2012). While the ecosystem response during heat 325 waves is linked to plant stress from excessively high temperatures and increased evaporative demand (i.e. higher vapour 326 pressure deficit), drought stress occurs when soil water supply can no longer meet the plant evaporative demand. The 327 former will lead to reduced carbon uptake through e.g. stomatal closure and disruptions in enzyme activity, the latter can 328 have direct impacts on carbon uptake by reducing stomatal and mesophyll conductance, the activity and concentrations of photosynthetic enzymes (Frank et al., 2015 and references therein). Apart from these almost instantaneous responses 329 330 additional lagged effects can further impact the carbon balance. If high temperatures were to occur in isolation we would 331 expect to observe a decrease in GPP. During the 2012/2013 heat waves in Australia, we observed a diurnal asymmetry in 332 GPP at all sites and in all measurement periods. This is expected in ecosystems that exert some degree of stomatal control 333 to avoid excessive reductions in water potential (e.g. in the afternoon), during higher atmospheric demand and when there is a reduced ability of the soil to supply this water to the roots because of lower matrix potentials and hydraulic 334 335 conductivity (Tuzet et al. 2003). Daily average carbon uptake at MW and TW was reduced by up to 32% and 40%, 336 respectively. At the TF site, however, daily averaged carbon uptake did not change significantly, and daytime carbon 337 uptake was significantly increased during both periods of the heat wave (see also Fig. 7). This can be explained partly by 338 the very dry conditions during the background period at this site, which could also have caused below average carbon 339 uptake, although comparing the site data against the long-term climatology confirmed an increased carbon uptake during 340 the heat wave (not shown). Although air temperatures clearly exceeded the ecosystem scale optimum of 18°C for carbon 341 uptake, and vapour pressure deficit exceeded values of 12 hPa, where stomatal closure can be expected at this site (van 342 Gorsel et al. 2013), increased incoming shortwave radiation (Tab. 2) more than compensated for these factors with 343 increased carbon uptake in this typically energy limited ecosystem during the heat wave. Overall, we have observed a 344 strong contrast between the water and energy limited ecosystems with the former (MW and TW) having strongly reduced GPP during heat waves and the latter (TF) having equal or slightly larger GPP. 345

346

Heat waves and drought not only affect photosynthesis but also have an impact on respiration (Frank et al., 2015). Increases in ER during the heat wave seem intuitive, given the exponential response of respiration to temperature (e.g. Richardson et al., 2006). Drought can also override the positive effect of warmer temperatures and lead to reduced respiration due to water limitations, as observed during the 2003 heat wave (Reichstein et al., 2007) or the 2011 spring drought (Wolf et al. 2013) in Europe. However, during the observed heat waves in Australia, increased air and soil temperatures led to significantly increased ecosystem respiration at all sites, indicating that the thermal response of respiration was undiminished despite soil moisture deficits.

While all sites remained carbon sinks during daytime hours in both heat wave periods, reduced carbon uptake in the 356 woodlands turned them to a net a source of carbon on a daily average. It can hence be concluded that increased ER combined with decreased or unchanged GPP likely turned large areas of Southern Australia from carbon sinks to sources. 357 358 Unlike the Mediterranean woodlands, the temperate woodlands recovered quickly after rain but the response of these 359 ecosystems to a short though intense heat wave indicates that future increases in the number, intensity and duration of heat 360 waves can potentially turn the woodlands into carbon sources, leading to a positive carbon-climate feedback. Heat waves 361 can also induce a transition from energy-limited to water-limited ecosystems (Zscheischler et al. 2015). Transitioning 362 toward water limitation, especially for energy-limited forests, can exacerbate the detrimental effects of extreme events. 363 Recurrent non-catastrophic heat stress can also lead to increased plant mortality, the impact of which would be more evident over longer timescales (McDowell et al., 2008) and as an increase in the frequency of fires (Hughes, 2003). 364 Similarly, legacy or carry-over effects of drought result in increased mortality and shifts in species composition during 365 subsequent years (van der Molen et al. 2011). Future climate change is likely to be accompanied by increased plant water 366 use efficiency due to elevated CO<sub>2</sub> (Keenan et al., 2013), which could lead to more drought and heat resilient plants, but 367 368 also to ecosystems with higher vegetation density and thus both higher water demands (Donohue et al., 2013, Ukkola et 369 al., 2015) and a greater susceptibility to large fires (Hughes, 2003). Furthermore, changes in the prevalence of drought will 370 affect forest carbon cycling and their feedbacks to the Earth's climate (Schlesinger et al. 2016). For Australia, there is 371 evidence that semi-arid ecosystems have a substantial influence on the global land carbon sink (Poulter et al 2014, 372 Ahlström et al. 2015). Due to their impact on the global carbon cycle, predicting the future influence of heat waves and 373 drought on the land sink of Australian woodlands thus remains a key research priority.

#### **4.3** The effect of intermittent precipitation during the heat wave

Intervening rain events led to differentiated responses in energy fluxes and lower air temperatures, but soil moisture content remained mostly low during HW2 (see section 3.1). Available energy was significantly lower (compared to BGH) during HW2 at MW but remained similar at TW and TF. At TF the latent heat flux in HW2 was still enhanced compared to BGH yet smaller than during HW1. Following rainfall the energy partitioning at the MW sites changed toward latent

- heat flux, with fractions similar to or larger than background conditions. This indicates that soil moisture feedbacks which inhibit warming of the lower atmosphere largely led to a return to standard conditions. At TW,  $\beta$  decreased to background values when precipitation occurred. While the magnitude returned to values similar to BGH, there was still a noticeable increase of  $\beta$  in the afternoon hours that was more pronounced than under average conditions. An increased fraction of energy going into the latent rather than the sensible heat during HW2 at the drier sites (MW and TW) does not only have important consequences on the soil moisture–temperature feedback but also on ameliorating vapour pressure deficit (Fig. 3) and reducing the atmospheric demand that acts as a stressor on plants (Sulman, et al. 2016).

During HW1, the time of maximum carbon uptake at the woodland sites was earlier in the morning than during BGH, and 388 we observed strongly reduced carbon uptake throughout the day. During HW2, however, the shift of maximum GPP 389 toward earlier hours of the day was less pronounced at MW and TW, thus daytime carbon uptake was not significantly 390 reduced. This was in response to the intermittent precipitation and lower temperatures, which led to a reduction in vapour 391 pressure deficit and increased soil water availability. Increased ER at all sites and during both HW periods was dominated 392 by warmer temperatures more than soil moisture limitations. Increased ER combined with decreased or unchanged GPP 393 likely turned large areas of Southern Australia from carbon sinks to sources, an effect that was reduced but not offset by 394 the intermittent precipitation.

When carbon losses exceed carbon gains over a long time period (e.g. through increased respiration) mortality can result 397 as a consequence of carbon starvation. Eamus et al. (2013) identified increased vapour pressure deficit as detrimental to 398 transpiration and net carbon uptake, finding that increased vapour pressure deficit is more detrimental than increased 399 temperatures alone - with or without the imposition of drought. A recent study by Sulman et al. (2016) confirmed that 400 episodes of elevated vapour pressure deficit could reduce carbon uptake regardless of changes in soil moisture. Here, all 401 ecosystems responded with increased carbon uptake to the precipitation events and the associated lower temperatures and 402 vapour pressure deficit. The improved meteorological conditions thus likely decreased the risk of mortality during HW2. 403 As heat waves increase in frequency, duration and intensity in the future (Trenberth et al. 2014), however, we expect a 404 decline in the ameliorating effects of intermittent rain events and an increased risk of mortality.

# 405 4.4 Comparisons to other heat waves (Europe, N America, China)

Anticyclonic conditions also caused the intense 2003 European heat wave (Black et al., 2004) as well as the even more intense and widespread heat wave that reached across Eastern Europe, including western Russia, Belarus, Estonia, Latvia, and Lithuania in 2010 (Dole et al. 2011). Less cloud cover and more clear sky conditions strongly increased incoming radiation and available energy during the European heat wave and drought in 2003 (Teuling et al., 2010), as well as during

the recent drought and heat in California (Wolf et al., in revision), in contrast to the current study. Teuling et al. (2010) 411 observed that surplus energy led to increases in both latent and sensible heat fluxes: over grassland, the energy was preferentially used to increase the latent heat flux, thereby decreasing  $\beta$ , whereas forest ecosystems generally had a 412 413 stronger increase in the sensible heat flux and an increase in  $\beta$  (Teuling et al. 2010, van Heerwaarden & Teuling 2014). 414 These results highlight the important ecosystem services provided by forests in the long-term, particularly considering the 415 increased prevalence of droughts and temperature extremes projected in the future (Trenberth et al. 2014). The situation in 416 our study was somewhat different in that soil water was only briefly limited in TF, where latent heat flux was mainly driven by temperature and vapour pressure deficit. After an intervening period of precipitation latent heat flux increased at 417 418 the drier sites (MW, TW) while sensible heat flux decreased or remained the same, potentially breaking the soil moisture-419 temperature feedback loop in Australia that maintained the heat wave in 2003 Europe. These findings highlight the 420 important role of Australian forest and woodland ecosystems in mitigating the effects of heat waves.

Stomatal control and reductions in GPP at the dry sites (MW and TW) were consistent and of similar magnitude with observations made during e.g. the 2003 European heat wave (Ciais et al. 2005), the 2010 European heat wave (Guerlet, 423 424 2013), the 2012 US drought (Wolf et al. 2016), and the 2013 heat wave and drought that affected large parts of Southern 425 China (Yuan et al. 2015). During these heat waves and droughts, carbon uptake was strongly reduced in general and 426 biosphere-atmosphere feedbacks from reduced vegetation activity further enhanced surface temperatures. This contrasts 427 with the wet site (TF), where local drought effects were observed only toward the end of the study. We found that the 428 response of carbon fluxes of Australian woodland (dry) ecosystems were similar to comparable heat waves on other 429 continents, whereas the detrimental effects of the heat wave were largely ameliorated in wet, energy-limited Australian 430 ecosystems.

Temperature anomalies during the 2012/2013 heat wave in Australia were less extreme ( $\leq -2 \sigma$ , Fig. 2) than during the 433 2010–2011 heat waves in Texas and Russia (-3  $\sigma$ ) and the 2003 European heat wave (> 2  $\sigma$ ) (Hansen et al., 2012; Bastos 434 et al., 2014), which resulted in smaller ecosystem responses than in Europe (Reichstein et al., 2007). However, this does 435 not imply that Australian heat waves are less severe than their Northern Hemisphere counterparts because background variability in climate, weather and ecosystem productivity are larger in Australia due to periodic synchronisation of El 436 437 Niño-Southern Oscillation, the Indian Ocean dipole and the state of the southern annular mode (Cleverly et al., 2016a). 438 When these climate modes are in phase, continental heat waves are strongly related to drought and reduced soil water content, although not to the same extent as in Europe during 2003 (Perkins et al., 2015). Nonetheless, responses of 439 440 Australian vegetation to heat waves and drought are consistent with vegetation responses elsewhere. For example during 441 the 2003 European heat wave, productivity in grasslands was most sensitive to heat and drought, while open shrublands 442 and evergreen broadleaf forests (like those in our study) were the least sensitive (Zhang et al., 2016). Two-thirds of the 443 productivity in Australia is due to CO2 uptake in non-woody ecosystems (Haverd et al., 2013), and it was indeed the semi-

- arid grasslands that produced the extraordinary CO<sub>2</sub> source strength during the drought and heat wave of January 2013
- (Cleverly et al., 2016b). Similarly, the semi-arid Mulga woodlands responded to the 2012/2013 heat wave with a large net
- source strength, increase in ecosystem respiration and afternoon depression in GPP (Cleverly et al., 2016b). We
- demonstrated in this study that *Eucalypt* forest and woodland ecosystems of southern Australia were more sensitive to
- heat waves if those ecosystems also experience moisture limitations.

# 449 **5** Conclusion

We have shown that extreme events such as the 'Angry Summer' 2012 / 2013 can alter the energy balance and therefore 450 451 dampen or amplify the event. During this event the woodland sites reduced latent heat flux by stomatal regulation in 452 response to the warm and dry atmospheric conditions. Stronger surface heating in the afternoons then led to an 453 amplification of the surface temperatures. Only the forest site AU-Tum had access to readily available soil water and 454 showed increased latent heat flux. The increased latent heat flux mitigated the effect of the heat wave but continuously 455 depleted the available soil water. The generally increased atmospheric and soil temperatures led to increased respiration 456 but unchanged net ecosystem productivity. The woodlands turned from carbon sinks into carbon sources and while the temperate woodlands recovered quickly after rain, the Mediterranean woodlands remained carbon sources throughout the 457 458 duration of the heat wave. This demonstrates that there is potential for positive carbon - climate feedbacks in response to 459 future extreme events, particularly if they increase in duration, intensity or frequency.

#### 461 Appendix A

We have used measurements of a reference period during the same season but one year after the heat wave 2012 / 2013 occurred. Ideally we would have used a climatology derived from observations but OzFlux is a relatively young flux tower network. The first two towers started in 2001 and even globally, very few flux towers have been measuring for more than 15 years, which is relatively short compared to typical climatology records of 30-years. To ensure the representativeness of our results we have therefore compared daily data against a climatology derived from BIOS2 output for the time period 1982-2013.

Table A1 shows the agreement between BIOS2 output for all sites and the time period the year 2013. Agreement was generally very good, even more so for the latent heat flux than for the carbon fluxes. Carbon fluxes, and more specifically respiration at the dry Mediterranean woodlands showed stronger disagreement. It is likely that this to some degree reflects night time issues with the eddy covariance method (e.g. van Gorsel, 2009) and with the partitioning of the measured fluxes. This may also be an indication that the model was underestimating drought-tolerance at these sites. The low modelled carbon uptake corresponded to periods of low soil water. There were long periods when the modelled soil water

- was below wilting point within the entire root zone of 4m. Underestimation could occur if roots were accessing deeper
- water, the wilting point parameter was too high or the modelled soil water was too low, relative to the wilting point.

Figure A1 shows that during HW1 the latent heat flux at the MW and TW sites was reduced. During HW2, precipitation and temporarily increased water availability brought the latent heat flux back to levels observed during BGH for the woodland sites. At the temperate forest, however, the latent heat flux strongly increased, particularly during HW1. Increasingly reduced soil water and lower temperatures reduced the effect during HW2.

- Figure A2 shows that carbon uptake was decreased at MW and TW during HW1 and similar to background conditions
- during HW2. At TF, the forest site, carbon uptake was increased. Respiration (Fig. A2b) was increased at all locations and
- during both heat wave periods.

#### 483 Acknowledgements

This work utilised data from the OzFlux network which is supported by the Australian Terrestrial Ecosystem Research 484 485 Network (TERN) (http://www.tern.org.au) and by grants funded by the Australian Research Council. We would like to 486 acknowledge the contributions Ray Leuning made to OzFlux and Au-Tum. Ray has been cofounder and leader of the OzFlux community and has been a great mentor to many in our network. We would also like to acknowledge the strong 487 488 leadership role that Helen Cleugh had over many years. The network would not be where it is without their input. VRD 489 and EP acknowledge the Education Investment fund and HIE for construction and maintenance or the AU-Cum tower. 490 The Australian Climate Change Science Program supported contributions by EvG and VH, SW was supported by the 491 European Commission's FP7 (Marie Curie International Outgoing Fellowship, grant 300083) and ETH Zurich. VRD acknowledges funding from a Ramón y Cajal fellowship RYC-2012-10970. NK acknowledges funding from The Royal 492 493 Society UK, Grant IE110132. We would further like to acknowledge the referees and their helpful comments that have 494 helped us to improve this manuscript.

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

# 835 Figures

836

Figure 1. Map indicating the locations of the OzFlux sites used in this study. The sites are grouped into three distinct climate and ecosystem types, indicated by red dots for Mediterranean woodlands (MW), light green dots for temperate woodlands (TW) and a dark green dot for the temperate forest (TF)

840