# Peer review of "Carbon uptake and water use in woodlands and forests in southern"

_Biogeosciences, 2016_

## Referee Comment (RC1) · Anonymous Referee #1 · 6 Jun 2016

This study details the the 'Angry Summer' of 2012/2013 in Australia from a FLUXNET perspective. I have a few sundry comments (listed below) but my main issue is one of narrative. I am missing some grand notion of what we've learned from this that we did not already know. We already knew, as an example, that extreme events alter the energy balance. We might have guessed that AU-Tum, as a forest with relatively deeper roots, would have a different LE response. Removing some of the verbosity might help polish the storyline, e.g., the site characteristics can be readily outsourced to a table in the appendix. Also, with your HW1 and the HW2 you are set up to talk about legacy (or memory) effects, but that word appears nowhere in the study. Finally,

your "Although all observed ecosystems remained carbon sinks through the duration of the heat wave" bit in the Conclusions is different that the expectation that extremes flip carbon status (sink to source). Using that as a framing piece would help the narrative I believe. In any event, as is I am at a loss as to what the main take-ways of this study are (other than different sites responded slightly differently, which is a given a priori).

Please use unique line numbering (do not reset to 1 every page). L5 is rather ambiguous here.

Re: "While climate change generally increases the sink strength of terrestrial ecosystems through carbon dioxide and nitrogen fertilisation" The authors need to rethink this. There have been several recent papers that read as an active debate on this assertion. See e.g., van der Sleen et al 2014 & Smith et al 2015. Plus there is more to climate change than C and N fertilization.

Re: "BIOS2" comes out of the blue here. In the abstract you mention CABLE?

Re: "Discussion" But you have not done anything with BIOS2 as of yet? Or if you have it is not clear what? Was it just to gapfill meteorological time series? In the appendix you have skill metrics for various fluxes but I cannot find a reference in the paper that these were ever used?

Figure 2: Why is precip different by panel? The same sites and times are shown in each one? Also, what of the background shading? I would put the legend outside of the plot as well.

Figure 3: This needs a legend for the symbols that is not in the caption.

Figure 4: What are "Flier points"?

Figure 5: (And this applies to the other figures as well.) Could you look into visually more distinct colors?

Figure 7: BHG? Why not show the differences wrt the background to declutter?

Table 1: Is it not Plains?

[Figure]

---

## Referee Comment (RC2) · Anonymous Referee #2 · 14 Jun 2016

Van Gorsel et al explore the sensitivity of Australian woodlands and forests to an extreme heat wave. Results are interesting but a number of minor steps could be taken to make the results less qualitative. The Discussion section should be expanded somewhat to create a more explicit comparison with other heat waves such as those studied by Ciais, Teuling, and many others. The Discussion section ends on a disappointing note with little in the way of synthesis of results to advance current knowledge. The following minor comments are designed to strengthen what I feel is an interesting manuscript that is of interest to the readership of Biogeosciences, but that needs to stake its claim to novelty.

[Figure]

Many qualitative statements can be avoided; for example what constitutes 'exceptional' on line 31? The abstract would be more powerful and citable with qualitativ3e statements instead of quantitative ones.

Define 'recover quickly' on page 3 line 16.

I'm confused, the end of the abstract says that CABLE was used but the end of the introduction says that BIOS2 was used.

I understand what is meant by 'relevant fluxes' but others outside the eddy covariance community might not.

Why is 'BGH' an acronym and how does it abbreviate 'reference period'? Avoid all acronyms that can be avoided. Please also state the actual name of the flux sites used rather than just the fluxnet acronym at first mention to provide a more complete description of the sites.

The sentence beginning line 16 page 4 includes the classifiers i, ii, iii, and then i again in a single sentence. Sometimes the common name for each species is in parentheses, and sometimes the scientific name is.

Occasional minor typographical issues like the space between 33 and m on page 5 line 3.

With respect to the GIMMS3g FPAR product, what product was used before this update?

More qualitative statements enter the results sections where they should be avoided at almost all costs. What is 'high' VPD and 'very low' soil water in section 3.1? See also: 'less pronounced', 'similar', 'decreased throughout the heat wave' (by how much?), and 'unusually high'. A nice statement follows 'unusually high': During HW1 they were generally more than 1.5-2 standard deviations ($\Leftrightarrow$) higher than during the same time during the 32 years mean from the background period (BGC). More passages should look like this.

Comma after 'Due to increased surface temperatures' on page 7. (Note 'moreso' following this passage doesn't say much. By how much?)

Superlatives like 'remarkably' and 'even' throughout the manuscript suggest surprise, but should be avoided. The reader knows the heat wave was big.

I don't understand 'daily latent heat flux (Fe)'. Is Fe a new abbreviation for latent heat flux on the daily time scale? In the discussion section on page 9 it is re-defined as F3. Wasn't the first usage of this common term somewhat sooner? Following the re-definition of Fe, the authors abbreviate sensible heat flux as 'Fh', then proceed to immediately not use this new definition in the next line. I recommend removing all abbreviations that are not necessary in this abbreviation-heavy manuscript.

On page 10 note that latent heat flux is also controlled by VPD in addition to soil heat flux and this stomatal control is discussed in the next paragraph.

Please define 'With temperatures clearly above an optimum temperature'. Plants can surprise. What is the optimum leaf temperature range for Eucalyptus? Also, how did phosotynthetically active leaf area potentially increase over such a short time period of the heat wave? I feel that this argument should be thought through a bit more. How do results agree or disagree with recent manuscripts by Poulter et al. (2014) and Ahlstroem (2015) on the role of dryland ecosystems in the global C budget? The end of the Discussion section is a bit vague and waves briefly at numerous diffuse threats yet doesn't synthesize results in any of these contexts.

The choice of red and green together in Figure 2 is a bit unfortunate. The legend says that some of the lines are meant to be blue but they appear green in my copy.

I'm not really sue what Figure 4 is telling us; I'm not accustomed to seeing things like incident radiation presented as a boxplot. Figure 5 is more useful.

Is Figure 6 just for one day? The smoothing/averaging treatment of Fig. 5 (and 7) would look nice here too.

Some thin vertical lines would help the multiple box plots be a bit more readable. It's hard to ascertain what corresponds to what.

A table of abbreviatons would help.

I feel that Ray Leuning should be included in the Acknowledgements section.

---

## Referee Comment (RC3) · Anonymous Referee #3 · 4 Jul 2016

This is a well written paper which adds to the growing body of science detailing the impacts of extreme heat events on terrestrial ecosystems. The authors introduce the story well indicating how the impacts of the extreme summer of 2013 in Australia will fit into the bigger global picture. They have then gone on to explain the finer nuances of just how the heat wave impacted on the southern Australian ecosystems during this period. In the Results and Discussion sections there is a reliance on Figure data rather than Tabular data and from this the presentation of the argument in the text is generally of a qualitative nature – this could be strengthened by putting some numbers in appropriate places. Another area where things fall a little short is in the Discussion

section where there is little pulling together of the Australian results with related studies from Europe, North America, China to generate a more holistic view of the impacts of these extreme heat events. Comments, suggestions for changes and areas that require some clarification: 1) As a comment: the number of sites in forested systems was n = 1. The behavior of this system AU-Tum, was somewhat different to the rest and was also different to the behavior of the European forest systems under this sort of extreme heat event. This wet site may be buffered from the sort of behavior that is usually seen until the soil moisture reserve that the forest can tap into has been reduced. This apparently did not happen during this heat wave until right at the end. The conclusion tells the story as it was for this site but not much can be generalized from this result unless this forest type remains in a wet environment in the future climate or if other temperature forests in southern Australia also have similar soil moisture reserves. 2) For explanation: The background period of hourly measurements that was used as a baseline comparison point for the study was quite short BGH (2/1/2014 – 6/1/2014) when compared with the period of the heat wave 1/1/2013 – 18/1/2013. It was quite understandable that more data was not used from 2014 if there was another event during January but what about January 2015 as this paper was submitted 9th May 2016? The results may not be altered but it would have been much more robust if a mean over 2 years was used or a longer period in 2015 was used rather than a 5 day window that was relatively warm (some sites z-score +1) and where the soil moisture was relatively low (some sites z-score -2) in 2014. 3) Climatology is used rather loosely in the text. Generally it refers to the computed climatology from BIOS2. If we go to the CSIRO website that describes BIOS2 (http://carbonwaterobservatory.csiro.au/bios2.html )we get: A library of programs for the download and treatment of inputs (gridded vegetation cover, meteorological data and parameters)... A weather generator for downscaling of meteorological data from daily to hourly. The BIOS2 modelling system computes the current and historical state of the major components of the carbon and water cycles for Australia at a spatial resolution of $0.05°$ latitude and longitude ($\sim$5 km), This could have been built into the paper more explicitly so that we understand what we are

getting – current and historical climate surfaces that are presumably calculated using Australian Bureau of Meteorology field data as primary data. In other places in the tables Climatology refers to the measured data from the flux towers. This needs to be specified more clearly. 4) Figure 2 I find confusing with the precipitation spread over the 3 panels. This would be better on its own as a 4th panel. 5) More needs to be put into the text on how close the various ecosystems were to switching from carbon sinks to sources. There were no numbers for sites to explain which sites switched to sources in Fig. 8 but what was provided indicated some MW site(s) (assuming the color code is brown) were sources in both HW1 and HW2 and some TW site(s) were sources in HW1. If this is correct then it is odd that this story was not developed as this would indicate some of these systems can be pushed over the edge by extreme heat events.

Listed details: P1/L33: BIOS2 to replace CABLE (throughout the text BIOS2 is used, CABLE is a part of this it would appear) P2/L9 : not be sustainable P3/L6: While greenhouse gas emissions generally (NOT climate change) P4/L4: Not really clear at this point how BGC and BGH differ. Spelling out that BGC is Modelled climate/fluxes and BGH are measured climate/fluxes would make it clearer. P4/L4: 2/1/2014 – 6/1/2014 Why was this reference chosen and not the mean from Jan 2014 and Jan 2015? P5/L1: the drier interior P5/L2: >30m as forests. P5/L2: Not sure what is intended here - None of this sites are montane? P5/L20: Is there a reference to ACCESS? P6/L6: Jupyter Notebooks. (otherwise it sounds like a type of hardware) P7/L28: Available energy – suggest providing a definition, this is not as well known as Bowen ratio P7/L29: at MW and TW sites during HW1 but was about the same for HW2 and the TF site. does this mean about the same for MW, TW and TF sites during HW2? P8/L16: to below background conditions in HW2 across the MW sites (?) P8/L18: For the TF site beta increased... P9/L10: pattern for associated P9/L15: heat wave in California.. P10/L4: or that there was little stomatal control of the latent heat flux at this site, This conflicts somewhat with what is written below: (L16) We observed a diurnal asymmetry in GPP at all sites and in all measurement periods. This is expected in ecosystems that exert some degree of stomatal control. (L30) With temperatures clearly above an optimum

temperature for carbon uptake and VPD exceeding values where stomatal closure can be expected at this site P10/L9: threshold of 0.4 Where does this threshold come from for the site? P10/L31: increased incoming shortwave Suggest using the z-scores for Fsd to indicate the difference to BGH and BGC P10/L32: increased photosynthetically active leaf area likely Is there an indication from MODIS that LAI has changed? You have reported MODIS LAI in Table 1. P11/L7: Figure 8 does not do this justice – what is needed is a Table at the site level. P11/L9: remained carbon sinks during both heat waves periods. This is why a table is needed - need to see that some sites are only just carbon sinks and some are sources. P11/L21: energy balance and therefore in concert with the current environmental conditions this may either mitigate.. (is this what was intended?) P11/L22: In the 'Angry Summer' event the woodland... P11/L28-30: Which of the sites in MW or TW are more at risk? P12/L16: HW1 the latent heat flux at the MW and TW sites was reduced even further. sense here is odd. We jump from the previous paragraph on carbon to this one on LE. ie. reduced even further means ? P12/L17:observed during BGH for the woodland sites. P16/L20: CO2 P18/L30: CO2 P21: F2 Would be better with a 4 panel that shows precipitation. It is confusing seeing rainfall for a TF site on a plot of MW etc. P22:L6 F3 stars denote soil water P23:L5 F4 energy imbalance is presented here and in the table and not discussed in the text. P25:L7 F7 dark green) heat at P26: F8 What does the color coding mean here? P27: FA1 What does the color coding mean here? to the BIOS2 climatology P28: T1 oC P29: T2 N.B. Here climatology means both observed and modelled.

---

## Author Response (AR1)

We would like to thank all reviewers for taking the time to read the manuscript and make many useful suggestions that we have used to make this a hopefully strongly improved version of the manuscript. We are grateful for the overall positive assessment and have incorporated the advice to improve the narrative, increased the quantitative nature of the text and improved the figures. The reviewer comments have in particular led to a reorganisation of the discussion section which should improve readability and better highlight the take away messages.

**Response to Anonymous Referee #1**

This study details the 'Angry Summer' of 2012/2013 in Australia from a FLUXNET perspective. I have a few sundry comments (listed below) but my main issue is one of narrative. I am missing some grand notion of what we've learned from this that we did not already know.

Australian ecosystems are generally known for their resilience to dry and hot conditions. Large-scale droughts and heat waves in Europe during 2003, in Canada during 2000 to 2003 and in the US during 2012 caused substantial reductions in summer carbon uptake, and vegetation-climate feedbacks were found to contribute to warmer temperatures. However, direct observations of the ecosystem response to large-scale extremes in Australia have been lacking so far and this manuscript provides new insights into such response based on direct flux measurements from several sites that has only recently become available. We will point this out much more clearly throughout the manuscript, will have a better emphasis on 'take away messages' and will change the abstract by adding "ecosystems known for their resilience towards hot and dry conditions." to "In this study we synthesised eddy-covariance measurements from seven woodlands and one forest site across three biogeographic regions in southern Australia. These observations were combined with model results from BIOS2 (Haverd et al., 2013) to investigate the effect of the summer heat wave on the carbon and water exchange of terrestrial ecosystems." in the abstract to reflect this earlier on in the manuscript.

We already knew, as an example, that extreme events alter the energy balance.
Yes, we do. However, our results provide novel insights into a range of heat wave responses across different ecosystems that have been previously unreported. Furthermore, as we point out in the discussion there is a fundamental difference in this Australian heat wave to what has previously been observed in Europe and US. In contrast to these other studies, incoming radiation and available energy were unaffected by the heat wave in Australia.

We might have guessed that AU-Tum, as a forest with relatively deeper roots, would have a different LE response.
Similar to reply 1) there have been no direct observations of ecosystem responses to large scale extremes in Australia. This is also true of studies characterising root distributions of these trees, thus we

cannot confirm (nor deny) the assumption that AU-Tum has trees with deeper roots than the other sites. However, it seems unlikely that the Mediterranean woodlands would not have equally deep or deeper roots because they are more strongly water limited than AU-Tum. This manuscript uses direct measurements to quantify this response beyond guessing and makes a contribution towards understanding the ecosystem response to heat waves in Asutralia.

Removing some of the verbosity might help polish the storyline, e.g., the site characteristics can be readily outsourced to a table in the appendix.

We will rework the storyline and cite characteristics for clarification. We do think, however, that describing the sites in detail is crucial for the interpretation of the results in this manuscript.

Also, with your HW1 and the HW2 you are set up to talk about legacy (or memory) effects, but that word appears nowhere in the study.

We chose to split the heat wave period into two parts to make it possible to see if the ecosystem sink / source behaviours would recover after a rain event. We will revise our argumentation for making this clearer in the revised version of the document (e.g. abstract).

The authors did not have any intent to investigate legacy effects which would require a different analysis altogether.

Finally, your "Although all observed ecosystems remained carbon sinks through the duration of the heat wave" bit in the Conclusions is different that the expectation that extremes flip carbon status (sink to source).

This aspect of the manuscript will change most during the revision. This sentence will be changed to "The woodlands turned from carbon sinks into carbon sources and while the temperate woodlands recovered quickly after rain, the Mediterranean woodlands remained carbon sources throughout the duration of the heat wave." We will adjust the conclusions and results/discussion accordingly. We will also contrast this finding to ecosystem responses in the European 2003 heat wave (Ciais et al. 2005 *Nature*; Bastos et al. 2014 *Biogeosciences*; van Heerwaarden et al. 2014 *Biogeosciences*), China (Bauweraerts et al. 2014 *Agricultural and Forest Meteorology*) and the 2013 summer drought in central Australia (Cleverly et al. 2016, *Agricultural and Forest Meteorology*).

Using that as a framing piece would help the narrative I believe. In any event, as is I am at a loss as to what the main take-ways of this study are (other than different sites responded slightly differently, which is a given a priori).

The authors don't think that previous to this study one could have safely guessed what the impact of extreme heat is on the different ecosystems across Australia and hope that the reviewer agrees that after the revisions made to the paper it is worth publishing this novel material that shows that the carbon sink of the the moisture limited, drought adapted woodlands is generally more vulnerable to heat extremes.

Please use unique line numbering (do not reset to 1 every page). L5 is rather ambiguous here.
Lines will be re-numbered throughout.

Re: "While climate change generally increases the sink strength of terrestrial ecosys- tems through carbon dioxide and nitrogen fertilisation" The authors need to rethink this. There have been several recent papers that read as an active debate on this assertion. See e.g., van der Sleen et al 2014 & Smith et al 2015. Plus there is more to climate change than C and N fertilization.
We will reduce the text and the references for carbon fertilisation as we feel discussion of this aspect distracts from the main topic, the impact of heat waves on the south Australian ecosystems, and is poorly supported by our data.

Re: "BIOS2" comes out of the blue here. In the abstract you mention CABLE?
Thanks, we will change CABLE to BIOS2 in the abstract.

Re: "Discussion" But you have not done anything with BIOS2 as of yet? Or if you have it is not clear what? Was it just to gapfill meteorological time series? In the appendix you have skill metrics for various fluxes but I cannot find a reference in the paper that these were ever used?
We refer to the climatology, derived from BIOS2 output in several places throughout chapter 3 (e.g. last sentence 3rd paragraph in 3.2.1 and last sentence of 3.2.1, last paragraph of 3.2.2).

Figure 2: Why is precip different by panel? The same sites and times are shown in each one? Also, what of the background shading? I would put the legend outside of the plot as well.
Graph will be changed also taking Referee3's comments into account

Figure 3: This needs a legend for the symbols that is not in the caption.
Thank you, adding a legend will improve readability.

Figure 4: What are "Flier points"?
Flier points are points that represent data extending beyond the whiskers (which represent the most extreme not outlier points). I.e. flier points are outliers. We will add this information to caption.

Figure 5: (And this applies to the other figures as well.) Could you look into visually more distinct colors?

The colour scheme is to a certain degree a matter of taste. The colours are either complementary colours or have a very distinct difference in grey scale to distinguish between them.

Figure 7: BHG? Why not show the differences wrt the background to declutter?
The reason for not showing the differences is that we would prefer not to lose the information on the magnitude of the fluxes.

Table 1: Is it not Plains?
It is indeed. Thank you.

**Response to Anonymous Referee #2**

Van Gorsel et al explore the sensitivity of Australian woodlands and forests to an extreme heat wave. Results are interesting but a number of minor steps could be taken to make the results less qualitative. The Discussion section should be expanded some- what to create a more explicit comparison with other heat waves such as those studied by Ciais, Teuling, and many others. The Discussion section ends on a disappointing note with little in the way of synthesis of results to advance current knowledge.

We will expand this section in the revised manuscript with the following references (amongst others). We will come the effects on energy and carbon fluxes in our study to other heat waves (at a minimum):

Ciais, P, Reichstein, M, Viovy, N, Granier, A, Ogee, J, Allard, V, Aubinet, M, Buchmann, N, Bernhofer, C, Carrara, A, Chevallier, F, De Noblet, N, Friend, AD, Friedlingstein, P, Grunwald, T, Heinesch, B, Keronen, P, Knohl, A, Krinner, G, Loustau, D, Manca, G, Matteucci, G, Miglietta, F, Ourcival, JM, Papale, D, Pilegaard, K, Rambal, S, Seufert, G, Soussana, JF, Sanz, MJ, Schulze, ED, Vesala, T, and Valentini, R: Europe-wide reduction in primary productivity caused by the heat and drought in 2003, Nature, 437, 529-533, 10.1038/nature03972, 2005.

Reichstein, M, Ciais, P, Papale, D, Valentini, R, Running, S, Viovy, N, Cramer, W, Granier, A, Ogee, J, Allard, V, Aubinet, M, Bernhofer, C, Buchmann, N, Carrara, A, Grunwald, T, Heimann, M, Heinesch, B, Knohl, A, Kutsch, W, Loustau, D, Manca, G, Matteucci, G, Miglietta, F, Ourcival, JM, Pilegaard, K, Pumpanen, J, Rambal, S, Schaphoff, S, Seufert, G, Soussana, JF, Sanz, MJ, Vesala, T, and Zhao, M: Reduction of ecosystem productivity and respiration during the European summer 2003 climate anomaly: a joint flux tower, remote sensing and modelling analysis, Glob. Change Biol., 13, 634-651, 10.1111/j.1365-2486.2006.01224.x, 2007.

Teuling, AJ, Seneviratne, SI, Stockli, R, Reichstein, M, Moors, E, Ciais, P, Luyssaert, S, van den Hurk, B, Ammann, C, Bernhofer, C, Dellwik, E, Gianelle, D, Gielen, B, Grunwald, T, Klumpp, K, Montagnani, L, Moureaux, C, Sottocornola, M, and Wohlfahrt, G: Contrasting response of European forest and grassland energy exchange to heatwaves, Nat. Geosci., 3, 722-727, 10.1038/ngeo950, 2010.

Trigo, RM, Garcia-Herrera, R, Diaz, J, Trigo, IF, and Valente, MA: How exceptional was the early August 2003 heatwave in France?, Geophys. Res. Lett., 32, 4, 10.1029/2005gl022410, 2005.

van Heerwaarden, CC, and Teuling, AJ: Disentangling the response of forest and grassland energy exchange to heatwaves under idealized land-atmosphere coupling, Biogeosciences, 11, 6159-6171, 10.5194/bg-11-6159-2014, 2014.

The following minor comments are designed to strengthen what I feel is an interesting manuscript that is of interest to the readership of Biogeosciences, but that needs to stake its claim to novelty.

Many qualitative statements can be avoided; for example what constitutes 'exceptional' on line 31?

It is difficult to explain 'exceptional' in the abstract but we quantify it quite clearly in the section "heat wave characterisation". Having said that we understand that this is a more general comment and will make changes in the abstract (and throughout the manuscript) to be more quantitative.

The abstract would be more powerful and citable with qualitative statements instead of quantitative ones.
We will add more quantification throughout abstract.

Define 'recover quickly' on page 3 line 16.
Will do

I'm confused, the end of the abstract says that CABLE was used but the end of the introduction says that BIOS2 was used.
Will change CABLE to BIOS2 in the abstract.

I understand what is meant by 'relevant fluxes' but others outside the eddy covariance community might not.
Thank you, we will change the term to "latent and sensible heat as well as carbon fluxes'

Why is 'BGH' an acronym and how does it abbreviate 'reference period'?
We will change the describing sentence to 'We used the hourly data of a background period (BGH)'

Avoid all acronyms that can be avoided. Please also state the actual name of the flux sites used rather than just the fluxnet acronym at first mention to provide a more complete description of the sites.

We will do so.

The sentence beginning line 16 page 4 includes the classifiers i, ii, iii, and then i again in a single sentence.
Will change, thank you.

Sometimes the common name for each species is in parentheses, and sometimes the scientific name is.
Will tidy up, thank you.

Occasional minor typographical issues like the space between 33 and m on page 5 line 3.
Will change and review the manuscript for corrections to symbols and text.

With respect to the GIMMS3g FPAR product, what product was used before this up-date?
We will change this section to read:
In this work, we updated BIOS2 to use the GIMMS3g FAPAR product (Zhu et al., 2013) instead of MODIS and AVHRR products (Haverd 2013b) for prescribed vegetation cover. The reference period used, BGC, is 1982-2013, the period over which remotely sensed data is available.

More qualitative statements enter the results sections where they should be avoided at almost all costs. What is 'high' VPD and 'very low' soil water in section 3.1? See also: 'less pronounced', 'similar', 'decreased throughout the heat wave' (by how much?), and 'unusually high'. A nice statement follows 'unusually high': During HW1 they were generally more than 1.5-2 standard deviations ($\Leftrightarrow$) higher than during the same time during the 32 years mean from the background period (BGC). More passages should look like this.
We will quantify these statements now throughout the results section.

Comma after 'Due to increased surface temperatures' on page 7. (Note 'moreso' following this passage doesn't say much. By how much?)
Comma will be added with quantified statements.

Superlatives like 'remarkably' and 'even' throughout the manuscript suggest surprise, but should be avoided.
We will mostly remove these words.

The reader knows the heat wave was big.
I don't understand 'daily latent heat flux (Fe)'. Is Fe a new abbreviation for latent heat flux on the daily time scale?
This new variable (Fe) was introduced in error and will be removed.

In the discussion section on page 9 it is re-defined as F3. Wasn't the first usage of this common term somewhat sooner? Following the re-definition of Fe, the authors abbreviate sensible heat flux as 'Fh', then proceed to immediately not use this new definition in the next line. I recommend removing all abbreviations that are not necessary in this abbreviation-heavy manuscript.

We will tidy this up

On page 10 note that latent heat flux is also controlled by VPD in addition to soil heat flux and this stomatal control is discussed in the next paragraph.Please define 'With temperatures clearly above an optimum temperature'. Plants can surprise. What is the optimum leaf temperature range for Eucalyptus?

We will more clearly point to the reference for the optimum temperature and provide ecosystem scale site specific numerical values.

Also, how did phosotynthetically active leaf area potentially increase over such a short time period of the heat wave? I feel that this argument should be thought through a bit more.

Agreed. We will remove this because it is not fully documented

How do results agree or disagree with recent manuscripts by Poulter et al. (2014) and Ahlstroem (2015) on the role of dryland ecosystems in the global C budget?

Poulter's study specifically focused on the response of ecosystems in the Southern Hemisphere to very wet, favourable conditions (in contrast to heat waves or droughts). However, when taken together with Zhao and Running (2010, *Science*), Ahlström (2014) and Cleverly et al. (2016 *Scientific Reports*), the extraordinary resilience of Australian ecosystems that dominates patterns of global productivity is a consequence of the extraordinary fluctuations in climate (from very wet to droughts and heat waves) to which the flora is adapted. This is in striking contrast to the 2003 heat wave in Europe, which was historically unprecedented (Schar et al. 2004 *Nature*). We will address in the Discussion how these results are consistent with previous findings on the role of dryland ecosystems on the global C budget.

The end of the Discussion section is a bit vague and waves briefly at numerous diffuse threats yet doesn't synthesize results in any of these contexts.

The end of the Discussion section will be revised to clarify and emphasise the difference of the responses of Australian ecosystems during the 2013 Angry Summer heat wave to observations made previously and elsewhere.

The choice of red and green together in Figure 2 is a bit unfortunate. The legend says that some of the lines are meant to be blue but they appear green in my copy.

We will change Figure 2 so that the blue lines are no longer applicable. We expect that providing stronger quantification in the text will improve the readability of the figures and mean that red/green is

less of a problem in this graph. In all other graphs there is a good grey scale separation through the lighter green.

I'm not really sue what Figure 4 is telling us; I'm not accustomed to seeing things like incident radiation presented as a boxplot. Figure 5 is more useful. Is Figure 6 just for one day? The smoothing/averaging treatment of Fig. 5 (and 7) would look nice here too.
Figure 4 shows the radiation balance and the partitioning of the energy terms during background conditions. We have added "Average" and a reference to Figure 5 to the caption of Figure 6 to emphasise that the lines are for the whole period of BGH, HW1 or HW2.

Some thin vertical lines would help the multiple box plots be a bit more readable. It's hard to ascertain what corresponds to what.
Will do

A table of abbreviatons would help.
Will add

I feel that Ray Leuning should be included in the Acknowledgements section.
Absolutely. We initially were going to dedicate the whole special issue to Ray (and therefore did not acknowledge him in this manuscript) but given current uncertainties around funding for a preface to the special issue we will add an acknowledgement to the manuscript. Furthermore, the introductory paper for the special issue (Beringer et al.) is dedicated to Ray Leuning.

Many thanks again for a very constructive review (which we will also acknowledge in the manuscript)!

**Response to Anonymous Referee #3**

In the Results and Discussion sections there is a reliance on Figure data rather than Tabular data and from this the presentation of the argument in the text is generally of a qualitative nature – this could be strengthened by putting some numbers in appropriate places.
We have tried to keep a balance between figures and tables. It is an excellent suggestion to weave numbers into the text to make the manuscript more quantitative– we will follow this suggestion in across the the Results and Discussion sections.

Another area where things fall a little short is in the Discussion section where there is little pulling together of the Australian results with related studies from Europe, North America, China to generate a more holistic view of the impacts of these extreme heat events.
We will expand the discussion to compare and contrast to heat waves in Europe, North America and China.

Comments, suggestions for changes and areas that re-quire some clarification: 1) As a comment: the number of sites in forested systems was n = 1. The behavior of this system AU-Tum, was somewhat different to the rest and was also different to the behavior of the European forest systems under this sort of extreme heat event. This wet site may be buffered from the sort of behavior that is usually seen until the soil moisture reserve that the forest can tap into has been reduced. This apparently did not happen during this heat wave until right at the end. The conclusion tells the story as it was for this site but not much can be generalized from this result unless this forest type remains in a wet environment in the future climate or if other temperature forests in southern Australia also have similar soil moisture reserves. Unfortunately there was no data available from another wet sclerophyll forest in Australia. We do think that our results are characteristic for wet sclerophyll forests but this may need to be confirmed when further sites become available. We will address these concerns in the site description of the revised manuscript.

2) For explanation: The background period of hourly measurements that was used as a baseline comparison point for the study was quite short BGH (2/1/2014 – 6/1/2014) when compared with the period of the heat wave 1/1/2013 – 18/1/2013. It was quite understandable that more data was not used from 2014 if there was another event during January but what about January 2015 as this paper was submitted 9th May 2016?
At the time of writing BIOS2 data was only available until end of 2013.
The results may not be altered but it would have been much more robust if a mean over 2 years was used or a longer period in 2015 was used rather than a 5 day window that was relatively warm (some sites z-score +1) and where the soil moisture was relatively low (some sites z-score -2) in 2014. 3) It was a tradeoff between increased time and number of sites. Including the 2015 period would have meant an overall reduction in sites as in 2015 some sites had gaps in the relevant period.

 3) Climatology is used rather loosely in the text. Generally it refers to the computed climatology from BIOS2.
We will make changes in the manuscript to refer to climatology only when talking about the computed climatology from BIOS2.

If we go to the CSIRO website that describes BIOS2 (http://carbonwaterobservatory.csiro.au/bios2.html ) we get: A library of programs for the download and treatment of inputs (gridded vegetation cover, meteorological data and parameters). A weather generator for downscaling of meteorological data from daily to hourly. The BIOS2 modelling system computes the current and historical state of the major components of the carbon and water cycles for Australia at a spatial resolution of 0.05∘ latitude and longitude (~5 km), This could have been built into the paper more explicitly so that we understand what we

are getting – current and historical climate surfaces that are presumably calculated using Australian Bureau of Meteorology field data as primary data.

We refer to the 2 papers in *Biogeosciences* led by Haverd in 2013 which give full detail of BIOS2. BIOS2 is doing much more than providing historical climate surfaces, which we will clarify in the model description of the methods.

In other places in the tables Climatology refers to the measured data from the flux towers. This needs to be specified more clearly.

We will make changes in the manuscript to refer to climatology only when talking about the computed climatology from BIOS2.

4) Figure 2 I find confusing with the precipitation spread over the 3 panels. This would be better on its own as a 4th panel.

We will do so

5) More needs to be put into the text on how close the various ecosystems were to switching from carbon sinks to sources. There were no numbers for sites to explain which sites switched to sources in Fig. 8 but what was provided indicated some MW site(s) (assuming the color code is brown) were sources in both HW1 and HW2 and some TW site(s) were sources in HW1. If this is correct then it is odd that this story was not developed as this would indicate some of these systems can be pushed over the edge by extreme heat events.

We would like to thank the reviewer for this valuable comment. We will extend the results section and discussion to include the point that we found a change in the number of hours during which the ecosystems acted as sources and sinks during the heat waves.

Listed details:
P1/L33: BIOS2 to replace CABLE (throughout the text BIOS2 is used, CABLE is a part of this it would appear)

Will do, thanks

P2/L9 : not be sustainable

Will do, thanks

P3/L6: While green- house gas emissions generally (NOT climate change)

thank you, we will change this paragraph

P4/L4: Not really clear at this point how BGC and BGH differ. Spelling out that BGC is Modelled climate/fluxes and BGH are measured climate/fluxes would make it clearer.

will do, thanks

P4/L4: 2/1/2014 – 6/1/2014 Why was this reference chosen and not the mean from Jan 2014 and Jan 2015?

We defined the reference period such that data would be available from all the sites. Unfortunately, 2015 did not have the same sites available.

P5/L1: the drier interior

Will do, thanks

P5/L2: >30m as forests.

Will do, thanks

P5/L2: Not sure what is intended here - None of this sites are montane?

We think it is worth pointing out the range of elevation. While 1260 m asl may not seem montane on other continents it puts the site into the temperate cool climate range.

P5/L20: Is there a reference to ACCESS?

Will do, thanks

P6/L6: Jupyter Notebooks. (otherwise it sounds like a type of hardware)

Will do, thanks

P7/L28: Available energy – suggest providing a definition, this is not as well known as Bowen ratio

Will do, thanks

P7/L29: at MW and TW sites during HW1 but was about the same for HW2 and the TF site. does this mean about the same for MW, TW and TF sites during HW2

we will clarify in the text

P8/L16: to below background conditions in HW2 across the MW sites (?)

Will add across the MW sites to the sentence, thanks

P8/L18: For the TF site beta increased…

Will change, thanks

P9/L10: pattern for associated

Will change

P9/L15: heat wave in California..

Will change

P10/L4: or that there was little stomatal control of the latent heat flux at this site, This conflicts some-
what with what is written below: (L16) We observed a diurnal asymmetry in GPP at all sites and in all
measurement periods. This is expected in ecosystems that exert some degree of stomatal control. (L30)
With temperatures clearly above an optimum C3 temperature for carbon uptake and VPD exceeding
values where stomatal closure can be expected at this site
That this could be due to limited stomatal control is a possibility, but we go on to exclude it.

P10/L9: threshold of 0.4 Where does this threshold come from for the site?
Will add (S. Zegelin, pers. comm.)

P10/L31: increased incoming shortwave Suggest using the z-scores for Fsd to indicate the difference to
BGH and BGC
We give absolute values on how Fsd differs between the heat wave periods and the background in table
2 but will add a reference to the table in the text.

 P10/L32: increased photosynthetically active leaf area likely Is there an indication from MODIS that
LAI has changed? You have reported MODIS LAI in Table 1.
MODIS data cannot be used to track changes over short time frames.  Because this analysis relies on
consistency of meteorological statistics over the analysis periods (thus we cannot choose longer analysis
periods), we will remove this reference. We do not know if LAI has changed due to the heat wave but
the fact that the temperate woodlands recovered after rain and in cooler temperatures indicates that there
may not have been permanent damage to the leafs.

P11/L7: Figure 8 does not do this justice – what is needed is a Table at the site level.
We will change fig.8 to provide the necessary numerical values and statistical inference of the compari-
son between background and heat wave values.

P11/L9: remained carbon sinks during both heat waves periods. This is why a table is needed - need to
see that some sites are only just carbon sinks and some are sources.
We will change fig.8 following the previous comment, which will identify and illustrate differences in
sink/source strength (as positive/negative NEP).

P11/L21: energy balance and therefore in con- cert with the current environmental conditions this may
either mitigate.. (is this what was intended?)
Will change to "dampen"

P11/L22: In the 'Angry Summer' event the woodland...
Will change to "during this event"

P11/L28-30: Which of the sites in MW or TW are more at risk?
While TW has restored its carbon sink function after the rain and under cooler temperatures MW has not. We will change text to address this point.

P12/L16: HW1 the latent heat flux at the MW and TW sites was reduced even further. sense here is odd. We jump from the previous paragraph on carbon to this one on LE. ie. reduced even further means ?
thanks, will clarify

P12/L17:observed during BGH for the woodland sites.
added, thanks

P16/L20: CO2
thanks, will change

P18/L30: CO2
thanks, will change

P21: F2 Would be better with a 4 panel that shows precipitation. It is confusing seeing rainfall for a TF site on a plot of MW etc.
thanks, will add a fourth panel.

P22:L6 F3 stars denote soil water
We agree that this was confusing. A legend will be added to the figure to clarify the symbols.

P23:L5 F4 energy imbalance is presented here and in the table and not discussed in the text.
Energy imbalance is a long-standing issue in eddy covariance research. However, the analysis presented in this study will not resolve the energy imbalance problem.

P25:L7 F7 dark green) heat at
Will change.

P26: F8 What does the color coding mean here?
Will add a reference to Figure 1 in the legend (i.e. "colours as in Fig. 1), thanks

P27: FA1 What does the color coding mean here? to the BIOS2 climatology

In the legend, colour coding will be defined relative to Fig. 1 and reference to BIOS2 climatology will be added, thanks

P28: T1 oC
thanks, will change

P29: T2 N.B. Here climatology means both observed and modelled.
thanks, will clarify

**Carbon uptake and water use in woodlands and forests in southern Australia during an extreme heat wave event in the 'Angry Summer' of 2012/2013.**

Eva van Gorsel[1], Sebastian Wolf[2], James Cleverly[3], Peter Isaac[1], Vanessa Haverd[1], Cäcilia Ewenz[4], Stefan Arndt[5], Jason Beringer[6], Víctor Resco de Dios[7], Bradley J. Evans[8], Anne Griebel,[5,9], Lindsay B. Hutley[10], Trevor Keenan[11], Natascha Kljun[12], Craig Macfarlane[13], Wayne S. Meyer[14], Ian McHugh[15], Elise Pendall[9], Suzanne M. Prober[13], Richard Silberstein[16]

1 CSIRO, Oceans and Atmosphere, Yarralumla, 2600, Australia

2 Department of Environmental Systems Science, ETH Zurich, 8092 Zurich, Switzerland

3 School of Life Sciences, University of Technology Sydney, Broadway, NSW, 2007, Australia

4 Airborne Research Australia, Flinders University, Salisbury South, SA, 5106, Australia

5 School of Ecosystem and Forest Sciences, The University of Melbourne, Richmond, 3121, Victoria, Australia

6 School of Earth and Environment (SEE), The University of Western Australia, Crawley, WA, 6009, Australia

7 Producció Vegetal i Ciència Forestal – Agrotecnio Centre, Universitat de Lleida, 25198, Leida, Spain

8 School of Life and Environmental Sciences, The University of Sydney, Sydney, 2015, Australia

9 Hawkesbury Institute for the Environment, Western Sydney University, Penrith, NSW 2570

10 School of Environment, Research Institute for the Environment and Livelihoods, Charles Darwin University, NT, Australia

11 Lawrence Berkeley National Lab., 1 Cyclotron Road, Berkeley CA, USA

12 Dept of Geography, College of Science, Swansea University, Singleton Park, Swansea, UK

13 CSIRO Land and Water, Private Bag 5, Floreat 6913, Western Australia

14 Environment Institute, The University of Adelaide, Adelaide SA 5 005, Australia

15 School of Earth, Atmosphere and Environment, Monash University, Clayton, 3800, Australia

16 Centre for Ecosystem Management, Edith Cowan University, School of Natural Sciences, Joondalup, WA, 6027, Australia

*Correspondence to*: E. van Gorsel (eva.vangorsel@csiro.au)

**Abstract.** As a result of climate change warmer temperatures are projected through the 21[st] century and are already increasing above modelled predictions. Apart from increases in the mean, warm/hot temperature extremes are expected to become more prevalent in the future, along with an increase in the frequency of droughts. It is crucial to better understand the response of terrestrial ecosystems to such temperature extremes for predicting land-surface feedbacks in a changing climate. While land-surface feedbacks in drought conditions and during heat waves have been reported from Europe and the US, direct observations of the impact of such extremes on the carbon and water cycles in Australia were lacking. During the 2012/2013 summer, Australia experienced a record-breaking heat wave with an exceptional spatial extent that lasted for several weeks. In this study we synthesized eddy-covariance measurements from seven woodlands and one forest site across three biogeographic regions in southern Australia. These observations were combined with model results from BIOS2 (Haverd et al., 2013) to investigate the effect of the summer heat wave on the carbon and water exchange of terrestrial ecosystems which are known for their resilience towards hot and dry conditions. We found that the water-limited woodlands and the energy-limited forest ecosystem responded differently to the heat wave. During the most intense part of the heat wave, the woodlands experienced decreased latent heat flux (up to 77% of background value), an increased Bowen ratio (154%) and a reduced carbon uptake (60%). At the same time the forest ecosystem showed increased latent heat flux (151%), reduced Bowen ratio (19%) and increased carbon uptake (112%). Higher temperatures caused increased ecosystem respiration at all sites (up to 139%). During daytime all ecosystems remained carbon sinks during the event but carbon uptake was reduced in magnitude. The number of hours during which the ecosystem acted as a carbon sink was also reduced which turned the woodlands into a carbon source. Precipitation increased after the most intense first part of the heat wave and cooler temperatures 
[revised manuscript text omitted]
, while in 2013 for both earlier and later time periods, data were not available for all sites at the time of data analysis. The reference period is shorter than the heat wave period because another significant heat wave event affected southeastern Australia later in January 2014 and at the end of 2013 not all sites had data available. Temperatures during the background reference period were also somewhat warmer than average climatology (Fig. 2). We therefore expect the relative severity of the effects of the heat wave to appear smaller than they otherwise would when compared against a climatological reference. To ensure the representativeness of our results, we also compared daily data against a climatology derived from daily BIOS2 (see below) output for the time period 1982-2013 (background climatology, BGC). BIOS2 results for the whole time period are only available as daily values.

**2.1 Sites**

We analysed data from seven southern Australian sites (Beringer et al., 2016) grouped in three distinct ecosystem and climate types: Mediterranean woodlands (MW), temperate woodlands (TW) and temperate forests (TF) (Fig. 1, Table 1).

MW sites included i) a coastal heath *Banksia* woodland (Gingin: AU-Gin) ii) a semi-arid eucalypt woodland dominated by Salmon gum *(Eucalyptus salmonophloia)*, with Gimlet *(E. salubrious)* and other eucalypts (Great Western Woodlands: AU-Gww); and iii) Calperum (AU-Cpr), which is situated in a semi-arid mallee ecosystem, which is characterised by an association of mallee eucalypts (*E. dumosa*, *E. incrassata*, *E. oleosa* and *E. socialis*) with spinifex hummocks (*Triodia basedowii*) (Sun et al. 2015, Meyer et al., 2015). TW sites are classified as dry sclerophyll and include i) Wombat (AU-Wom), which is a secondary re-growth of Messmate Stringybark *(E. oblique)*, Narrow-Leaved Peppermint *(E. radiate)* and Candlebark *(E. rubida)* (Candlebark). ii) Whroo (AU-Whr) is a box woodland mainly composed of Grey Box *(E. microcarpa)* and Yellow Gum *(E. leucoxylon)*. Smaller numbers of Ironbark *(E. sideroxylon)* and Golden Wattle *(Acacia pycnantha)* also occur on site. iv) In Cumberland Plains (AU-Cum) the canopy is dominated by Gum-topped Box *(E. moluccana)* and Red Ironbark *(E. fibrosa)*, which host an expanding population of mistletoe *(Amyema miquelii)*. Temperate

[revised manuscript text omitted]

cooling was observed in the eastern states after the 8th of January, but in the meantime a second high pressure system moved into the Bight, starting a second wave of record breaking heat across the continent. The heat wave finally ended on the 19th of January, when southerly winds brought cooler air masses to southern Australia.

Figure 3 shows the meteorological conditions at the sites during the heat wave. Maximum temperatures as high as 46.26°C were accompanied by vapour pressure deficits up to 9.70 kPa. The soil water fraction was as low as 0.02 in MW but increased to 0.05 and 0.4 at AU-Gin and AU-Gww respectively after synoptic rainfalls around the 12th of January. The same, but less pronounced, was also the case for the TW sites where soil water fractions increased from 0.10 to 0.18 after rain. At the TF site, Au-Tum, soil water decreased throughout the heat wave (HW) from 0.26 to 0.19. Due to intermittent precipitation events we analysed two parts of the heat wave separately: heat wave period 1 (HW1, 1st–9th of January 2013) was characterised by very little precipitation (2 mm over all sites) and low soil water. During heat wave period 2 (HW2, 10th–18th of January 2013) precipitation occurred at most sites (12th–15th January 2013) and resulted in increased soil water and lower temperatures than during HW1.

During HW1 temperatures were generally more than 1.5-2 standard deviations (σ) higher than during the same time during the 32 years mean from the background period (BGC). At AU-Tum and AU-Gww z-scores even exceeded +2σ. During HW2 all sites showed lower z-scores for temperatures but they were still more than +1σ higher than average temperatures. The background period BGH, against which we compare the hourly data of the heat wave, was also warmer than average conditions during the past 30 years but for most sites these z-scores were well below 1.

Z-values indicate that soil water content was unusually low for the time of year. It was mostly more than -1σ below average, except at AU-Gww where soil water content was higher than average during HW2. All sites except the AU-Gin and AU-Gww had a lower z-score for soil water during HW2 than HW1, indicating relatively drier conditions with respect to the BIOS2 derived climatology despite the presence of rainfall during HW2. The background period BGH was generally less dry than the heat wave periods, one noteworthy exception being AU-Tum, which had very dry conditions (-2σ) during BGC in early January 2014. The z-scores indicate that high temperatures were more unusual than low soil water during HW1. HW2 was both hot and dry.

**3.2 Ecosystem response to dry and hot conditions**

**3.2.1 Energy Exchange**

Incoming and reflected short-wave radiation were only significantly increased by 70 Wm$^{-2}$ and 3 Wm$^{-2}$ respectively in the energy limited ecosystem AU-Tum during the first period of the heat wave (Fig. 4, Table 2). Otherwise they remained approximately the same as BGH values except at the MW sites where they were significantly reduced (by -62 Wm$^{-2}$) during HW2 (Table 2). The relatively short duration of the extreme heat wave did not result in changes in albedo (not shown). A warmer atmosphere and potentially increased cloud cover led to a 38 Wm$^{-2}$ increase in longwave downward radiation in

Western Australia. Due to increased surface temperatures, longwave radiation emitted at the land surface was significantly increased at all sites for both heat wave periods (28 Wm$^{-2}$ on average), though more so during HW1 (41 Wm$^{-2}$ on average). Significantly reduced net radiation was only measured at MW sites during HW2 (-35 Wm$^{-2}$). At all other sites, net radiation was approximately the same during HW1, HW2 and BGH. Available energy (not shown), the energy available to the turbulent heat fluxes, was significantly reduced at MW and TW sites during HW1 (by 25Wm$^{-2}$ and 24Wm$^{-2}$ respectively) but was about the same for HW2. It was also about the same during HW1 and HW2 at the TF site.

Figure 5 demonstrates how remarkably different the energy partitioning was at MW, TW and TF sites, as we would expect given the wide climatological and biogeographic differences (Beringer et al., 2016). While similar fractions of energy went into latent and sensible heat at the TF site, more energy was directed into sensible heat at TW sites. This energy flux partitioning towards sensible heat was more pronounced at MW sites, where both the mean and the variability of latent heat flux were very small due to severe water limitations. Most of the available energy was transferred as sensible heat and hence contributed to the warming of the atmosphere which was also observed for BGH.

During HW1, the variability and the daily average of the generally small latent heat flux at the MW sites (38 Wm$^{-2}$) were further reduced by -12 Wm$^{-2}$ (see also Table 2). During HW2, precipitation and temporarily increased water availability brought the latent heat flux back to levels observed during BGH. The latent heat flux did not change significantly at TW sites during the HWs compared to BGH conditions. At TF, however, the latent heat flux increased by 52 and 14 Wm$^{-2}$ during HW1 and HW2 respectively. This was partly due to the very dry conditions in the background period BGH, but daily latent heat flux was also increased compared to the climatology (BGC, Fig. A2), particularly during HW1.

With values exceeding 7, the observed ratio of sensible to latent heat, the Bowen ratio ($\beta$, Bowen, 1926), was very large in the Mediterranean woodlands (Fig. 6). Typical values for $\beta$ reach 6 for semi-arid to desert areas (e.g. Oliver, 1987, Beringer and Tapper, 2000). During the heat wave these values were larger than 10. With rainfall and an increased latent heat flux, $\beta$ decreased to below background conditions in HW2 (6.4) across the MW sites. TW, $\beta$ was higher than background values (reaching maximum values of 4.0 and 2.8 respectively) but decreased to similar values during HW2 (3.0). For the TF site, $\beta$ was lower (0.7 and 0.8 for HW1 and HW2 respectively) than during the background period (1.0). It increased steadily in the morning and declined towards the evening and was quite symmetric, while in TW $\beta$ increased strongly in the afternoon during the heat waves. This increase of $\beta$ towards the afternoon hours was observed in MW during all time periods (including BGH).

Measured daily latent heat fluxes and $\beta$ were consistent with flux climatology derived from BIOS2 during the background (BGH) (Fig. A1).

**3.2.2 Carbon Exchange**

Patterns of carbon fluxes were similar to between-site patterns of energy fluxes (Fig. 7, note differences in y-axes). All sites showed that maximum carbon uptake (GPP) occurred in the morning, decreased throughout the afternoon, and mostly increased again in the late afternoon. NEP followed the diurnal course of GPP, with the offset related to total ecosystem respiration (ER). ER increased with temperature and reached a maximum in the early afternoon (not shown). Maximum NEP at MW decreased from 4.16 $\mu mol m^{-2} s^{-1}$ during background conditions to 2.2 $\mu mol m^{-2} s^{-1}$ in HW1 and 3.3 $\mu mol m^{-2} s^{-1}$ 
[revised manuscript text omitted]

[Figure]

Figure 2. z-scores for temperature and soil water across flux tower sites. Solid red stars denote temperature and non filled red stars denote soil water scores during the background period (BGH) compared to the climatological background BGC. Scores for temperature (T) and soil water (Sw) for HW1 and HW2 compared to the same time periods in the years 1982-2013 are shown for HW1 (1/1/2013–9/1/2013) by filled dots and for HW2 (10/1/2013–18/1/2013) by non-filled dots.

[Figure]

Figure 3. Time series of daily maximum temperature (T max, top panel), daily maximum vapour pressure deficit (VPD max), soil water and precipitation. The legend is given in the top panel. Precipitation (P) is given as the average of the daily cumulated precipitation of the sites and displayed for each biome.Shaded areas in the background indicate the time periods HW1 and HW2.

[Figure]

Figure 4. Box plot of energy fluxes for Mediterranean woodlands (MW, red), temperate woodlands (TW, light green) and temperate forests (TF, dark green). Energy fluxes are incoming shortwave radiation (Fsd), reflected shortwave radiation (Fsu), downward longwave radiation (Fld), emitted longwave radiation (Flu), net radiation (Fn), latent heat flux (Fe), sensible heat flux (Fh), ground heat flux (Fg) and energy imbalance ($\varepsilon$) during the background period BGH (2/1/2014 – 6/1/2014). The box extends from the lower to upper quartile values of the data, with a line at the median. The mean value is indicated with a dot. The whiskers extend from the box to show the range of the data. Flier points (outliers, blue dots) are those past the end of the whiskers.

[Figure]

Figure 5. Diurnal course of net radiation (Fn, light amber), sensible (Fh, red) and latent (Fe, blue) heat at the Mediterranean woodlands (MW, top row), the temperate woodlands (TW, middle row) and the temperate forest (TF, lowest row) for the background period BGH (2/1/2014 – 6/1/2014), and the first and second period of the heat wave (HW1 (1/1/2013–9/1/2013) , HW2 (10/1/2013–18/1/2013)). Filled areas indicate the range of smoothed ±1 standard deviation, average mean values are indicated by symbols.

[Figure]

Figure 6. Average daytime Bowen ratio measured over Mediterranean woodlands (MW, left panel), the temperate woodlands (TW, middle panel) and the temperate forest (TF, right panel) for BGH (green line), HW1 (light amber) and HW2 (dark amber).

Figure 7. Diurnal course of net ecosystem productivity (NEP, light green) and gross primary productivity (GPP, dark green) at the Mediterranean woodlands (MW, top row), the temperate woodlands (TW, middle row) and the temperate forest (TF, lowest row) for the background period (BHG), and the first and second period of the heat wave (HW1, HW2). Filled areas indicate the range of smoothed ±1 standard deviation values, average mean values are indicated by red symbols. Background GPP values (dark grey) and NEP values (light grey) are also plotted in HW1 and HW2 to allow for easier comparison.

[Figure]

Figure 8. Boxplot of daytime values (9:00-16:00 local standard time) of gross primary productivity (GPP), ecosystem respiration (ER) and net ecosystem productivity (NEP) for the background period (BGH) and the first and second period of the heat wave (HW1, HW2). Daytime average values (DTA) are given below boxes and symbols indicate that they are significantly different from the background period (o) or not (x). Daily averages (0:00-23:00, local standard time) and their significance are also given. Colours as in Fig. 1.

[Figure]

Figure A1. Left panel: Boxplot of the ratio of observed latent heat (Fe(obs)) to the BIOS2 climatology of the latent heat flux (Fe(BGC)) during the first and second period of the heat wave (HW1, HW2). Right panel: same as left but for the Bowen ratio. Colours as in Fig. 1.

[Figure]

Figure A2. Left panel: Boxplot of the ratio of observed gross primary productivity (GPP(obs)) to the climatology of GPP (GPP(BGC)) during the first and second period of the heat wave (HW1, HW2). Right panel: same as left but for the ER. Colours as in Fig. 1.

**Tables**

**Table 1.** List of OzFlux sites used in this study, abbreviations and site information. MW stands for Mediterranean woodlands, TW for temperate woodlands and TF for temperate forest. MAT and MAP are the mean annual temperature and precipitation for the years 1982-2013 (BIOS2).

| ID | Site Name | Latitude (deg) | Longitude (deg) | Eleva-tion (m) | MAT (°C) | MAP (mm) | LAI Modis ($m^2/m^2$) | Tree Height (m) | Biome |
|---|---|---|---|---|---|---|---|---|---|
| AU-Gin | Gingin | -31.375 | 115.714 | 51 | 18.4 | 681 | 0.9 | 7 | MW |
| AU-Gww | Great Western Woodlands | -30.192 | 120.654 | 450 | 18.7 | 396 | 0.4 | 25 | MW |
| AU-Cpr | Calperum | -34.004 | 140.588 | 67 | 17.0 | 265 | 0.5 | 4 | MW |
| AU-Wom | Wombat | -37.422 | 144.094 | 702 | 11.4 | 936 | 4.1 | 25 | TW |
| AU-Whr | Whroo | -36.673 | 145.029 | 155 | 14.6 | 533 | 0.9 | 30 | TW |
| AU-Cum | Cumberland Plains | -33.613 | 150.723 | 33 | 17.6 | 818 | 1.3 | 23 | TW |
| AU-Tum | Tumbarumba | -35.657 | 148.152 | 1260 | 9.8 | 1417 | 4.1 | 40 | TF |

**Table 2.** Statistics of radiation and energy exchange for the ecosystems Mediterranean Woodlands (MW), Temperate Woodlands (TW) and Temperate Forests (TF) and the variables flux of shortwave downward radiation ($F_{sd}$), shortwave upward radiation($F_{su}$), longwave downward radiation ($F_{ld}$), longwave upward radiation ($F_{lu}$), net radiation ($F_n$), latent heat ($F_e$), sensible heat ($F_h$), ground heat ($F_g$) and the energy imbalance ($\varepsilon$). Where values during the two periods of the heat wave ($\Delta$HW1 and $\Delta$HW2) differ significantly from the background (BGH) (P < 0.1) this is indicated by bold fonts.

| | MW | TW | TF | MW | TW | TF | MW | TW | TF | MW | TW | TF | MW | TW | TF | MW | TW | TF | MW | TW | TF | MW | TW | TF | MW | TW | TF |
|---|---|---|---|---|---|---|---|---|---|---|---|---|---|---|---|---|---|---|---|---|---|---|---|---|---|---|---|
| | $F_{sd}$ | | | $F_{su}$ | | | $F_{ld}$ | | | $F_{lu}$ | | | $F_n$ | | | $F_e$ | | | $F_h$ | | | $F_g$ | | | $\varepsilon$ | | |
| BGH | 335 | 264 | 293 | 49 | 30 | 31 | 350 | 383 | 323 | 439 | 444 | 387 | 197 | 175 | 197 | 38 | 63 | 103 | 130 | 90 | 55 | 5 | 1 | 0 | 24 | 21 | 39 |
| ΔHW1 | -10 | 4 | **70** | -1 | 0 | **3** | **38** | 9 | 0 | **49** | **33** | **41** | -19 | -20 | 26 | **-12** | 2 | **52** | -5 | **30** | -8 | 6 | **4** | **5** | **-8** | **-56** | **-23** |
| ΔHW2 | **-62** | 20 | 16 | **-10** | 2 | 14 | **36** | 2 | **-13** | **19** | **16** | **12** | **-35** | 3 | -8 | -3 | -2 | **14** | **-29** | -4 | -8 | -5 | 1 | 0 | 1 | 8 | **-14** |

**Table A1.** Parameters of robust linear model fit between observations and BIOS2 output for all sites, the variables latent heat flux (Fe), gross primary productivity (GPP), ecosystem respiration (ER) and the time interval 1/1/2013 – 31/12/2013.

| variable | ID | coeff | stderr | [95% conf. int.] | | RMSE | $r^2$ |
|---|---|---|---|---|---|---|---|
| $F_e$ | | | | | | | |
| | AU-Gin | 0.9998 | 5.87e-06 | 1.000 | 1.000 | 27.52 | 0.44 |
| | AU-Gww | 0.9285 | 0.0032 | 0.867 | 0.990 | 23.13 | 0.58 |
| | AU-Cpr | 0.7464 | 0.032 | 0.684 | 0.809 | 15.75 | 0.41 |
| | AU-Wom | 1.1253 | 0.017 | 1.093 | 1.158 | 18.55 | 0.89 |
| | AU-Whr | 0.8917 | 0.031 | 0.830 | 0.953 | 23.66 | 0.42 |
| | AU-Cum | 1.0385 | 0.020 | 0.999 | 1.078 | 28.75 | 0.53 |
| | AU-Tum | 0.8121 | 0.014 | 0.784 | 0.840 | 36.61 | 0.65 |
| GPP | | | | | | | |
| | AU-Gin | 1.0570 | 0.035 | 0.988 | 1.126 | 1.91 | 0.42 |
| | AU-Gww | 0.3496 | 0.031 | 0.290 | 0.410 | 0.95 | 0.15 |
| | AU-Cpr | 0.2806 | 0.021 | 0.240 | 0.321 | 1.06 | 0.10 |
| | AU-Wom | 1.0176 | 0.015 | 0.988 | 1.047 | 1.53 | 0.81 |
| | AU-Whr | 1.0194 | 0.037 | 0.947 | 1.092 | 2.32 | 0.38 |
| | AU-Cum | 1.8764 | 0.037 | 1.803 | 1.949 | 2.81 | 0.48 |
| | AU-Tum | 0.6667 | 0.006 | 0.655 | 0.679 | 3.20 | 0.88 |
| ER | | | | | | | |
| | AU-Gin | 1.1687 | 0.043 | 1.084 | 1.253 | 2.20 | 0.13 |
| | AU-Gww | 0.4483 | 0.015 | 0.420 | 0.477 | 0.66 | 0.31 |
| | AU-Cpr | 0.3492 | 0.013 | 0.324 | 0.374 | 0.73 | 0.01 |

| | | | | | | | |
|---|---|---|---|---|---|---|---|
| AU-Wom | 1.2682 | 0.035 | 1.199 | 1.337 | 2.56 | 0.48 |
| AU-Whr | 1.4227 | 0.039 | 1.347 | 1.499 | 1.87 | 0.14 |
| AU-Cum | 2.0017 | 0.031 | 1.941 | 2.063 | 2.62 | 0.66 |
| AU-Tum | 0.8770 | 0.013 | 0.852 | 0.902 | 1.69 | 0.77 |